# Long-term fluxes of carbonyl sulfide and their seasonality and interannual variability in a boreal forest

Timo Vesala[1,2,3], Kukka-Maaria Kohonen[1], Linda M.J. Kooijmans[4], Arnaud P. Praplan[5], Lenka Foltýnová[6], Pasi Kolari[1], Markku Kulmala[1], Jaana Bäck[2], David Nelson[7], Dan Yakir[8], Mark Zahniser[7], Ivan Mammarella[1]

[1]Institute for Atmospheric and Earth System Research / Physics, University of Helsinki, Helsinki, Finland
[2]Institute for Atmospheric and Earth System Research / Forest Sciences, University of Helsinki, Helsinki, Finland
[3]Yugra State University, 628012, Khanty-Mansiysk, Russia
[4]Meteorology and Air Quality, Wageningen University & Research, Wageningen, The Netherlands
[5]Finnish Meteorological Institute, Helsinki, Finland
[6]Global Change Research Institute, Czech Academy of Sciences, Brno, Czech Republic
[7]Aerodyne Research Inc., Billerica, MA, USA
[8]Weizmann Institute of Science, Rehovot, Israel

*Correspondence to*: Kukka-Maaria Kohonen (kukka-maaria.kohonen@helsinki.fi)

**Abstract.** The seasonality and interannual variability of terrestrial carbonyl sulfide (COS) fluxes are poorly constrained. We present the first easy-to-use parameterization for net COS forest sink based on the longest eddy covariance record from a boreal pine forest, covering 32 months over 5 years. Fluxes from hourly to yearly scales are reported, with the aim of revealing controlling factors and the level of interannual variability. The parameterization is based on the photosynthetically active radiation, vapor pressure deficit, air temperature, and leaf area index. Wavelet analysis of the ecosystem fluxes confirmed earlier findings from branch-level fluxes at the same site and revealed a 3-hour lag between COS uptake and air temperature maxima in the daily scale, while no lag between radiation and COS flux was found. The spring recovery of the flux after the winter dormancy period was mostly governed by air temperature, and the onset of the uptake varied by 2 weeks. For the first time, we report a significant reduction of ecosystem-scale COS uptake under large water vapor pressure deficit in summer. The maximum monthly and weekly median COS uptake varied 26 and 20 % between years, respectively. The timing of the latter varied by 6 weeks. The fraction of the nocturnal uptake remained below 21 % of the total COS uptake. We observed the growing season (April–August) average net flux of COS totaling -58.0 gS ha$^{-1}$ with 37 % interannual variability. The long-term flux observations were scaled up to evergreen needleleaf forests (ENFs) in the whole boreal region by the Simple Biosphere Model Version 4 (SiB4). The observations were closely simulated by using SiB4 meteorological drivers and phenology. The total COS uptake by boreal ENF was in line with a missing COS sink at high latitudes pointed out in earlier studies.

# 1 Introduction

During the last decade, carbonyl sulfide (COS) has attracted attention among the scientific community investigating the carbon cycle. Although the actual contribution of the exchange rate of COS between the biosphere and atmosphere to the ecosystem carbon balance is extremely small, COS has been proposed to provide a new insight into carbon dioxide ($CO_2$) exchange as a promising tracer (proxy) for the gross carbon uptake of plants (e.g., Sandoval-Soto et al., 2005; Asaf et al., 2013; Whelan et al., 2018). The COS exchange can also provide valuable insight into the dynamics and estimates of the stomatal conductance regulating plant gas exchange and evapotranspiration (Wehr et al., 2017; Kooijmans et al., 2019; Stoy et al., 2019). Leaves and soil are the largest sink for COS (e.g., Kesselmeier et al., 1999; Whelan et al., 2018) and the net biosphere–atmosphere exchange of this trace gas has potential impact on the climate (e.g., Crutzen, 1976). COS affects climate through ozone chemistry and aerosol production besides its direct warming effect. Brühl et al. (2012) found that the consequent net cooling from aerosols is cancelled out by the warming.

The terrestrial plant COS uptake estimate has a broad range from 400 to 1360 Gg S $y^{-1}$ (Campbell et al., 2017; Remaud et al., 2021; Hu et al., 2021). Recently the inversion modeling study by Ma et al. (2021) pointed to missing sources in the tropics and missing sinks at high latitudes. The factors controlling the COS flux (FCOS) temporal variatiability are partly unknown, although the dependence on temperature and light are rather well understood during the growing season (Commane et al., 2015; Wehr et al., 2017; Kooijmans et al., 2019). The prerequisite for fundamental understanding of the dynamics of the COS budget is *in situ* net ecosystem-scale flux observations with high time resolution by the eddy covariance (EC) method. Nevertheless, these direct flux measurements are still scarce. We report here the multiyear COS surface net flux for 32 months over 5 years from a boreal forest. This provides not only the opportunity to analyze the seasonality and interannual variability of the flux but also to create a representative parameterization presented here for the first time for the multiyear flux. The longest reported EC flux record before this study is from Wehr et al. (2017) from May through October for 2 years. They focused on average diurnal cycles and seasonality based on biweekly means in analyses of canopy stomatal conductance. The data presented here are unique in their location, being from the boreal region (mature Scots pine stand at Hyytiälä/SMEAR II station; Hari & Kulmala, 2005). The earlier reported EC records were collected from the Mediterranean (Asaf et al., 2013; Wohlfahrt et al., 2018; Yang et al., 2018; Spielmann et al., 2019) and temperate broadleaf ecosystems / arable land (Billesbach et al., 2014; Maseyk et al., 2014; Commane et al., 2015; Wehr et al., 2017; Spielmann et al., 2019).

Our objective is to report the net FCOSs and analyze them from hourly to yearly scales together with environmental conditions and other controlling factors, while paying attention to the seasonality and interannual variability. We hypothesize that the long-term FCOS can be described by a simple semi-empirical parameterization that employs radiation, air temperature ($T_a$), humidity, and leaf area index (LAI). We test the hypothesis on our long time series. We analyze the dependence of FCOS on environmental factors both by using multivariate and univariate linear regressions, combining

factors, and by applying wavelet analysis. COS balances with shares between day and nocturnal uptake are provided. We focus on the net ecosystem FCOS without partitioning into soil and canopy components since the net exchange is one of the main constraints on the atmospheric concentration, and especially because the multiyear dynamics of FCOS have not been previously studied. For canopy and soil COS exchange in the studied pine forest, see Kooijmans et al. (2017, 2019) and Sun et al. (2018), respectively. Finally, we demonstrate the value of the long-term COS time series by using the upscaled parameterized FCOSs to evaluate FCOS simulations by the Simple Biosphere Model Version 4 (SiB4). We reflect on the results with the net $CO_2$ exchange but investigations of the gross carbon uptake are beyond the scope of this study.

## 2 Materials and methods

### 2.1 Site description

Measurements were made at the SMEAR II station, a boreal coniferous forest in Hyytiälä in southern Finland (61°51′ N, 24°17′ E, 181 m ASL) during the years 2013–2017. The site was dominated by Scots pine (*Pinus sylvestris* L.) with some Norway spruce (*Picea abies* (L.) Karst.) and deciduous trees (e.g., *Betula* sp., *Populus tremula*, *Sorbus aucuparia*) within at least a 150 m radius from the measurement mast. The tree density is ~1170 ha$^{-1}$ (Ilvesniemi et al. 2009), and the canopy height increased from approximately 18 m to 20 m during the measurement years. The Scots pine stand was established in 1962 (Hari & Kulmala, 2005). The long-term annual mean precipitation and the annual mean temperature are 711 mm and 3.5 °C, respectively (Pirinen et al. 2012).

### 2.2 Eddy covariance measurements and flux processing

EC measurements were made at 23 m height using an ultrasonic anemometer (Solent Research HS1199, Gill Instruments, Lymington, UK) to measure three wind components at 10 Hz frequency and an Aerodyne quantum cascade laser spectrometer (QCLS, Aerodyne Research, Billerica, MA, USA), which measured COS, $CO_2$, water vapor ($H_2O$), and since 2015 also carbon monoxide (CO) mole fractions also at 10 Hz. The gas flow rate was approximately 10 standard liters per minute. Background measurements of high-purity nitrogen were made every 30 min for 26 s to remove background spectral structures (Kooijmans et al., 2016). EC data were processed using the EddyUH software (Mammarella et al., 2016) following the recommendations given in Kohonen et al. (2020). Raw data were despiked based on the maximum difference allowed between two subsequent data points, two-dimensional coordinate rotation was used to rotate the coordinate frame, and turbulent fluctuations were determined from linear detrending. Lag times of COS, carbon monoxide (CO), and carbon dioxide ($CO_2$) were determined from the maximum cross-covariance of $CO_2$ with vertical wind speed (*w*), while the lag time of water vapor ($H_2O$) was determined from the maximum cross-covariance of $H_2O$ with *w*. High-frequency spectral corrections were calculated according to Mammarella et al. (2009), so that the response time of $CO_2$ was also used for COS and CO spectral corrections. Low-frequency correction was done according to Rannik & Vesala (1999).

Quality screening was applied to the data by tests for flux stationarity (≤ 0.3) and limits for kurtosis (1 < Ku < 8) and skewness (−2 < Sk < 2). In addition, only 100 spikes were allowed for each 30 min period and the second wind rotation angle was not allowed to exceed 10 degrees. Friction velocity ($u_*$) filtering with a threshold of 0.3 m/s was applied to exclude time periods with low turbulence. Finally, fluxes were gap-filled using the function

$$FCOS = a*I/(I+b)+c*D+d \qquad (1)$$

where $I$ is photosynthetic photon flux density, $D$ is vapor presssure deficit (VPD), and $a$, $b$, $c$, and $d$ are fitting parameters (Kohonen et al., 2020).

The uncertainty of the cumulative flux was estimated using a bootstrap method. This was done because the uncertainty of single 30 min fluxes consists of both random and systematic uncertainties, as well as the uncertainty of the gap-filling function that cannot be mixed with flux measurement uncertainties. In the bootstrap method, we assumed a 20% total uncertainty that mostly consists of processing uncertainty (Kohonen et al., 2020), and a synthetic data set was randomly sampled from the data set (including the 20 % uncertainty) 10 000 times. The synthetic data sets consisted of an amount of data points equal to those in the original data set, but as the samples were drawn at random, some data points may be included several times while other points may not be included at all. The overall uncertainty was calculated as the difference to the 95th percentile of the 10 000 bootstrap sample sums. The FCOS record comprises 32 months over 5 years, of which 51 % is gap-filled, representing 23 152 measured 30 min fluxes, which provides the opportunity to observe and analyze the interannual variability.

## 2.3 Environmental variables and the commencement of the carbonyl sulfide uptake period

$T_a$ and relative humidity (RH) were measured at 16.8 and 33.6 m heights and an average of the two heights was assumed to best represent $T_a$ and RH at 23 m height, where the EC measurements were made. $T_a$ was measured by Pt100 sensors and RH calculated from the $H_2O$ mole fraction measured with a LI-COR LI-840 infrared light absorption analyzer and a Pt100 sensor. Photosynthetically active radiation (PAR) from 400 to 700 nm was measured above the canopy by a LI-COR LI-190SZ quantum sensor. Soil temperature ($T_s$) in the A horizon (2–5 cm in the mineral soil) was determined as the mean of five locations representative of the forest floor, with data measured using Philips KTY81-110 temperature sensors. Volumetric soil water content (SWC) in the A horizon (2–6 cm depth) was also calculated as the mean of five different locations, with data measured using a Campbell TDR100 time-domain reflectometer. Optical LAI was calculated from continuous measurements of PAR with 8 sensors at 0.6 m height near the flux tower and one sensor above the forest using inverse Beer-Lambert equation parameterized separately for direct beam and diffuse radiation. Wintertime LAI (October through March) was interpolated. The vapor pressure deficit (VPD) was calculated as the difference between the saturation water vapor pressure ($e_s$) and the actual water vapor pressure ($e_a$) as

$$VPD = e_s - e_a \tag{2}$$

where

$$e_s = 0.618 \exp\left(\frac{17.27\, T_a}{T_a + 237.3}\right) \tag{3}$$

$$e_a = \frac{RH\, e_s}{100} \tag{4}$$

where $T_a$ is air temperature and RH relative humidity of air, both calculated as an average of measurements done at 16.8 m and 33.6 m heights, representing the flux measurement height of 23 m.

Commencement of the growing season was determined, following the recommendations of Suni et al. (2003a), by a moving average $T_a$ with a 5-day window (with a threshold of 3.3 °C indicating the start of photosynthetic uptake) and by a $T_a$

dependent variable $S$ that describes the stage of physiological development, introduced by Pelkonen & Hari (1980):

$$S_i = S_{i-1} + S_t \tag{5}$$

where

$$S_t = \frac{100}{1 + 100\, a^{-(T_a - S/c)}} - \frac{100}{1 + 100\, a^{(T_a - S/c)}} \tag{6}$$

In Eq. 6, $a = 2$ and $c = 600$ are fitted constants as in Suni et al. (2003a) and Pelkonen & Hari (1980), and Eqs 5 and 6 were

140 solved with the Euler method with a half-hour time step dt. For simplicity, we scaled $S$ by $c$ as in Suni et al. (2003a). The threshold for the start of the growing season by this method was $S/c > 1.81$, as in Suni et al. (2003a), and $S$ at $i = 0$ was defined as $S_0 = 0$. The $S$ parameter has been used in ecosystem research especially to determine the spring recovery (Suni et al., 2003; Pelkonen & Hari 1980) and biosphere modeling in JSBACH (Mäkelä et al., 2016; Mäkelä et al., 2019). This is very useful especially at high latitudes due to variating spring conditions. When temperatures may change by over 10°C in

one day and vary both below and above 0°C, the more traditional heat sum is not a good approximation for phenology. The $S$ parameter takes into account the temperature history and sub zero temperatures, which the traditional heat sum does not.

Although the progress of COS uptake in the spring is a gradual process, we get a better insight into its evolution and its yearly variations if we define a threshold for the onset of COS uptake. For the commencement of the significant COS uptake period, we used the date when the midday FCOS permanently (for at least 5 consecutive days) fell below 30 % of the median

of the minumum FCOS from all years (maximum uptake) (see Table S1). Daytime was defined as periods when the solar elevation angle was greater than zero.

## 2.4 Multivariate linear regression analysis

Multivariate linear regression analysis was used to find the dependencies of FCOS on various environmental factors: VPD, $T_a$, PAR, RH, $T_s$, SWC, net radiation ($R_n$), and atmospheric COS mixing ratio. All the computations were done in R (R Core Team, 2019) using base functions, and the variance inflation factor (VIF) was computed using the vif function within package car (Fox & Weisberg, 2011). Only measured (non-gap-filled) FCOS data was used for the analysis so that minimum 50 % of data had to exist to calculate e.g. daily averages.

We used VPD and $T_a$ within the same linear regression model, despite knowing that VPD is obtained using $T_a$. However, VPD is computed as an exponential function of $T_a$, which technically does not break the assumption of linear independence of explanatory variables within the regression model. Nonetheless, there is naturally some multicollinearity among almost all environmental factors, which is necessary to treat. The VIF was used to see whether the multicollinearity among explanatory variables within each regression model is of an acceptable level to not brake the assumption of the linear regression analysis. The VIF quantifies the severity of multicollinearity in an ordinary least-squares regression analysis. It provides an index that measures how much the variance (the square of the estimate's standard deviation) of an estimated regression coefficient is increased because of collinearity. VIF < 10 is a commonly used cutoff for acceptable multicollinearity. All the models which included more than one variable were tested with VIF, which was in all cases less than 10. Two variables showing the same physical quantity were never used in the same model ($T_a$ and $T_s$, PAR and $R_n$, or RH and VPD).

## 2.5 Wavelet coherence analysis

Magnitude-squared wavelet coherence was computed using MATLAB's (MATLAB R2019a) wcoherence function, which uses the analytic Morlet wavelet (Grinsted et al., 2004; Lau & Weng, 1995). As input to the function, fluxes were first quality screened and gap-filled. For longer measurement gaps (e.g., winter), fluxes were forced to zero. Thus, for longer data gaps the wavelet coherence is also zero. Environmental variables (PAR, $T_a$, and VPD) were averaged from 1 min to 30 min values to match the time stamp of the flux data.

## 2.6 Parameterization of carbonyl sulfide fluxes

FCOS was parameterized using PAR, $T_a$, VPD, and total LAI data over the 32 months on a daily scale. Since the ecosystem COS uptake is dominated by the canopy uptake (70% at minimum according to Sun et al., 2018) and is process-wise very close to the $CO_2$ uptake, we formulated the parameterization as follows:

$$FCOS = FPAR * FS * FVPD * FLAI \qquad (7)$$

where

$$FPAR = \frac{a * PAR}{PAR + b} \qquad (8)$$

$$FS = \frac{1}{1 + \exp(c * S)} \qquad (9)$$

$$FVPD = \frac{d}{1 + \sqrt{VPD}} \qquad (10)$$

$$FLAI = \frac{1 - \exp(-e * LAI)}{e} \qquad (11)$$

where $a$, $b$, $c$, $d$, and $e$ are fitting parameters and four functions are simplified from the corresponding dependencies of the CO$_2$ uptake on PAR and $T_a$ (S being a function of $T_a$; see Text S2) (according to Mäkelä et al., 2008), VPD (Dewar et al., 2018), and LAI (Peltoniemi et al., 2015). Parameters $a = -341.81$, $b = 1000$, $c = -0.77$, and $d = 1.03$ were optimized using MATLAB's fminsearchbnd function to find the smallest root mean square error of the parameterized FCOS against the measured (non-gap-filled) FCOS. The fminsearchbnd function finds the minimum of a constrained multivariable function using a derivative-free method (D'Errico, 2021). Parameters were given upper and lower limits of $a \in [5,500]$, $b \in [10,1000]$, $c \in [-3,3]$, $d \in [1,5]$. It is to be noted that parameter $b$ is at its upper limit. However, increasing the limit (until infinity) for parameter $b$ only resulted in a higher value for parameter $a$, without significantly improving the overall fit. Parameter $e = 0.18$ was fixed before optimizing the other parameters according to a previous study by Peltoniemi et al., (2015), since parameter $e$ is related to ecosystem phenology specific to the site, and is believed not to be gas dependent. Although recognizing that other more complex formulas with more fitting parameters could provide better correspondence with observations, we desired to keep the parameterization simple for the sake of generic process description: here FPAR describes the stomatal response to PAR and includes all other light-dependent processes, FS the phenology of biochemical reactions, FVPD the stomatal regulation, and FLAI the amount of foliage and canopy light penetration. The VPD response was based on Dewar et al. (2018), who showed that the FVPD function is predicted theoretically by a variety of stomatal optimization models, and explains observed stomatal responses better than the empirical Ball-Berry model (Ball et al., 1987).

**2.7 Boreal region carbonyl sulfide fluxes**

We scaled up the FCOS parameterization to the whole boreal region to evaluate FCOS simulations by SiB4. SiB4 (Haynes et al., 2019) is a continuation of the SiB3 model in which COS exchange was implemented by Berry et al. (2013). One of the added capabilities of SiB4 that was not present in SiB3 is that it simulates fluxes of multiple plant functional types (PFTs) in

a grid cell and allows selection of fluxes from a single PFT. This is beneficial for the analysis in this study where we want to compare with observation-based data from evergreen needleleaf forests (ENFs).


The COS module of SiB4 was recently updated with the COS soil exchange model of Ogee et al. (2016) and the standard COS mole fraction of 500 ppt was replaced by COS mole fraction fields that vary in space and time, including seasonal and diurnal variability (Kooijmans et al., 2021). The meteorological data that drive SiB4 are from the Modern-Era Retrospective analysis for Research and Applications (MERRA2), available from 1980 onward (Gelaro et al., 2017). The simulations of

SiB4 are preceded by a spin-up from 1850 to 1979 to initialize the carbon pools using the climatological average of available MERRA2 data (Smith et al., 2020). We ran SiB4 with 3-hourly output globally and for the analysis we selected grid cells (at $0.5° × 0.5°$ resolution) where ENFs cover more than 30 % of the land area in the northern hemisphere. The information on areal coverage of different PFTs was retrieved from MODIS data (Lawrence & Chase, 2007). Finally, only the fluxes representing ENF were selected.


To obtain COS biosphere fluxes for the whole boreal region based on the FCOS observations in Hyytiälä we calibrated the SiB4 PAR, LAI, VPD and $T_a$ for the grid cell where Hyytiälä is located against observations. The obtained calibration is shown in Fig. S10. The in-situ LAI is the all-sided leaf area index, while SiB4 LAI is projected leaf area index, which explains the large difference between the two LAI data. We then applied the parameterization represented in Eqs. 7-11 to the

whole boreal region (based on the ENF grid cell selection described in the previous paragraph) using the SiB4 meteorological and phenological data.

## 3 Results and discussion

### 3.1 Environmental conditions

PAR ranged from 0 to 1900 µmol m$^{-2}$s$^{-1}$ with the highest amount of radiation reached usually in June (Fig. 1). May 2017 and

July 2014 had higher monthly median PAR (285 and 303 µmol m$^{-2}$s$^{-1}$, respectively) than other years (Fig. S1). $T_a$ varied from minimum -28°C in January 2016 to maximum 29°C in July 2014 while the mean $T_a$ of the whole measurement period was 4.9°C. Highest monthly median $T_a$ was reached in July 2014 (19.5°C). May-June 2013 and July 2014 were warmer compared to other years while April 2017 was colder than usual (monthly median $T_a$ 0.4°C). $T_s$ had more moderate variation from 2.7°C to 17.5°C and with an average over the whole period of 5.8°C. VPD ranged from 0 to 2.7 kPa with the highest

VPD reached during May-July, depending on the year. May 2013 and 2016 were dryer than usual with a monthly average VPD of 0.6 kPa. July 2014 and August 2015 were also unusually dry with monthly median VPD 0.7 and 0.5 kPa, respectively, but were not considered as a drought as the SWC remained at a normal level (well above 0.1 m$^3$m$^{-3}$). SWC did not vary much between different years during the growing seasons (April-October) but August 2015 and 2016 had higher SWC than other years. The minimum SWC (0.1 m$^3$m$^{-3}$) was usually reached in August-September, while it was highest right

after snow melt in April (monthly median 0.3 m³ m⁻³ ). LAI varied seasonally between 5 and 7 m² m⁻² with maximum LAI
reached in August in all years.

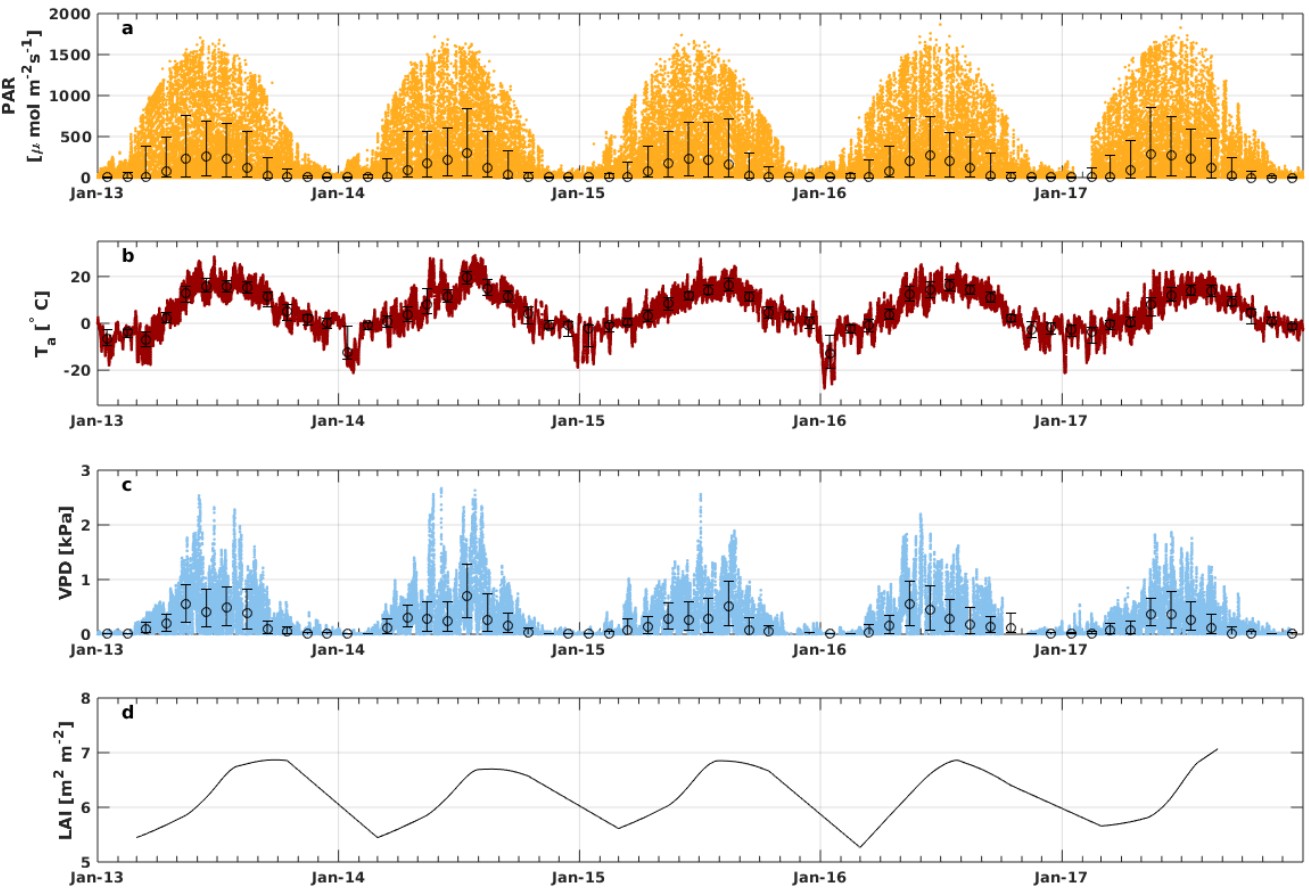

**Figure 1. Timeseries of PAR (a), Tₐ (b), VPD (c) and all-sided LAI (d). Small dots in a-c represent 30 min values while circles**
**represent the monthly medians and errorbars show their 25ᵗʰ and 75ᵗʰ percentiles. LAI are daily smoothed values.**

## 3.2 Seasonality of FCOS and its interannual variability

FCOS (COS uptake indicated by a negative sign) showed a pronounced seasonal cycle (Fig. 2a–d, Fig. S2) with the most
negative flux in summer (June–August), which was expected as PAR, $T_a$ and $T_s$ also had clear seasonal cycles (Figs 1a and b,
Fig. S3a).

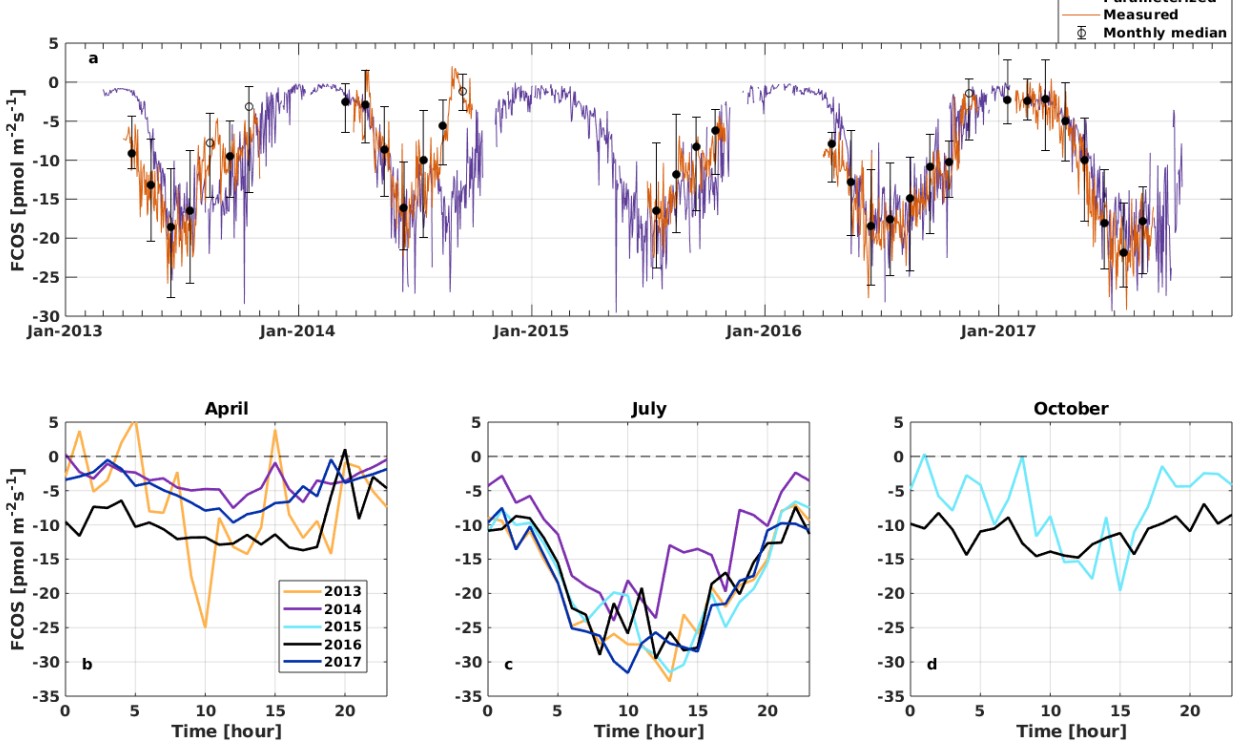

Figure 2. Daily average net carbonyl sulfide flux (FCOS) time series (a) and median diurnal variation of FCOS in April (b), July (c), and October (d). A negative sign means COS uptake. Measured data (orange line in (a)) show daily gap-filled averages (see Sect. 2.2) and black circles represent the monthly medians. Filled circles mean that more than 20 % of data was measured, whereas empty circles indicate less than 20 % of measured data for that month. Whiskers show the 25th and 75th percentiles. The purple line in (a) is the parameterized daily FCOS (see Eq. 1). Plots b–d represent measured (non-gap-filled) data only. No data are available for 2015 in April and for 2013, 2014, and 2017 in October. Parameterized values are shown for 5 full years.

In April, years 2013 and 2016 represented higher uptake than years 2014 and 2017 (Fig. 2b). The significant COS uptake started on 15 April in 2013, based on a 3-day midday flux moving average being persistently below 30 % of the summer minimum (ca. $-10$ pmol m$^{-2}$ s$^{-1}$) (Table S1), and before 1 April (before measurements started) in 2016. The corresponding date was 28 April both in 2014 and 2017. Suni et al. (2003a) and Sevanto et al. (2006) studied the same forest stand as here and found that the spring recovery of the $CO_2$ exchange is controlled by $T_a$ to a large extent, whereas the soil water availability is not a limiting factor. Snow melt has been also used as a good proxy indicator for the start of the carbon uptake (Pulliainen et al., 2017) in the boreal region. Suni et al. (2003a) defined the photosynthetically active period as when the $CO_2$ flux reached 20 % of the maximum summer $CO_2$ flux. They found that the days on which the daily average or a moving average $T_a$ with a 5-day window exceeded 4.0 or 3.3 °C, respectively, coincided accurately with the commencement day of the photosynthetically active period. We analyzed the dependence of $CO_2$ flux on $T_a$ for the period of this study and no similar corresponding relationship was found for $CO_2$ nor for FCOS. When the median $T_a$ of May in 2014 and 2017 was low (below 8 °C), the FCOS remained above $-9$ pmol m$^{-2}$ s$^{-1}$, whereas it was ca. $-13$ pmol m$^{-2}$ s$^{-1}$ in 2013 and 2016 when the

median $T_a$ was ca. 13 °C (Table S2, Fig. S1). During spring (April–May) the higher VPD was positively correlated with the COS uptake, but this resulted from higher $T_a$, which was also positively correlated (see Figs 3b and c). The heat sum (sum of daily average temperatures that are above 0 °C) did not influence the commencement of significant COS uptake in the spring (Fig. S4).

The COS uptake in July 2014 was at a significantly lower level in the afternoon than in other years (Fig. 2c). $T_a$ and VPD having maximum values in the afternoon were larger in July 2014 than in any other year (Fig. S1). This led to the most pronounced asymmetry in the uptake between the morning and afternoon in July 2014. The monthly median COS uptake was 54 % smaller in July 2014 compared with that in 2017 when the uptake was the largest (Table S2). The forest acted as a COS sink during nighttime throughout the measurement period (see Fig. 6). Previous measurements have found both nocturnal soil uptake (Sun et al., 2018), but also uptake by foliage, as suggested by the stomatal conductance being non-zero throughout the night, indicating incomplete stomatal closure during nighttime (Kooijmans et al., 2017; 2019). Mosses and other cryptogams can also significantly take up COS during the night when the soil is wet (Rastogi et al., 2018). Nighttime soil uptake in Hyytiälä was ca. -3 pmol m$^{-2}$s$^{-1}$ (Sun et al., 2018) while the ecosystem scale nighttime uptake in our study is ca. -10 pmol m$^{-2}$s$^{-1}$. However, we did not measure the contribution from cryptogams. The nighttime COS uptake in Hyytiälä is thus likely a combination of soil and moss uptake, but also has a larger contribution from the canopy (Kooijmans et al., 2017). The average nocturnal FCOS varied between -5 to -12 pmol m$^{-2}$ s$^{-1}$ throughout the year. Compared with July, the diurnal variation in FCOS was much smaller in April and October, especially in 2016 (Figs 2b –d). The flux was close to zero but still indicated small uptake during winter months when data were available in November 2016–March 2017 (monthly median between $-2.4$ and $-1.5$ pmol m$^{-2}$ s$^{-1}$) (Fig. 2a, Table S2). Suni et al. (2003b) found that light is the determining factor for the cessation of the growing season for the same stand. For the parameterized flux in Fig. 2a, see Sect. 3.4.

The lowest monthly median FCOS (highest uptake) was $-21.9$ pmol m$^{-2}$ s$^{-1}$ in July 2017 (Table S2). In June 2014 the monthly FCOS was $-16.1$ pmol m$^{-2}$ s$^{-1}$, which is the highest flux from the months of the biggest uptake in each year. The lowest weekly FCOS was $-23.9$ pmol m$^{-2}$ s$^{-1}$ in week 29 (20–26 July) in 2017, and the highest was $-19.2$ pmol m$^{-2}$ s$^{-1}$ in week 25 in 2014 (16–22 July), from the weeks of the biggest uptake in each year. Thus, the interannual variability in the maximum uptake is 26 and 20 % in the monthly and weekly medians, respectively. The variability in the timing of the maximum weekly uptake was 6 weeks, occurring in week 23 in 2016 and week 29 in 2017 (Table S3).

**3.3 Carbonyl sulfide flux relationship on light, air temperature, and vapor pressure deficit**

Fig. 3 presents FCOS as a function of PAR, $T_a$, and VPD for April–May and June–August. Responses to $T_a$ and VPD were filtered to only include data with PAR > 500 µmol m$^{-2}$ s$^{-1}$ to avoid including radiation related correlation, since VPD and $T_a$ are highly intercorrelated with PAR. For the responses without separation between spring and summer periods and without

filtering by low PAR, see Fig. S5. The fluxes showed a clear relationship with PAR (Figs 3a and d), even though the COS biochemical reactions and CA activity are light independent and FCOS responds to light mainly due to the light response of

stomatal conductance (Kooijmans et al., 2019). Besides CA, carboxylating enzymes ribulose-l,5-bisphosphate-carboxylase (RuBisCO) and phosphoenolpyruvate-carboxylase (PEP-Co), that are light dependent, contribute to COS metabolism (Protoschill-Krebs and Kesselmeier, 1992). However, it is not possible to separate between these processes from EC flux measurements and it has been shown that the main enzyme contributing to COS uptake in plants is CA (Protoschill-Krebs et al., 1996). Thus, we do not expect the role of PEP-Co and RuBisCO light-dependency to exceed that of stomatal regulation.

When $T_a$ was high (Fig. 3e), in theory favoring the uptake due to enhanced biochemical reactions in the mesophyll, the simultaneous occurrence of high VPD (Fig. 3f) limited the uptake due to smaller stomatal conductance values (see Kooijmans et al., 2019). The fluxes tended to saturate as a function of $T_a$ above 10 °C in spring and increase (uptake decrease) above 15 °C in summer (Figs 3b and e, respectively). At 10 °C in spring FCOS was ca. -15 pmol m$^{-2}$ s$^{-1}$, whereas in summer ca. -25 pmol m$^{-2}$ s$^{-1}$. These flux values correspond to VPD of ca. 1 kPa in spring and summer, respectively. The

non-zero value in the dark represents the ecosystem nocturnal COS uptake (Fig. 3a). The response curve saturated at high PAR values in the summer to approximately twice the level of those in spring. The saturation occurred above the PAR value of about 500 µmol m$^{-2}$ s$^{-1}$. Correspondence with VPD resembled that on $T_a$ (Figs 3c and f). The fluxes tended to increase (uptake decrease) when VPD was above 0.8 kPa.

We analyzed the relationship between FCOS and environmental factors using multivariate and univariate linear regressions combining VPD, relative humidity (RH), net radiation ($R_n$) or PAR, $T_a$ or $T_s$, SWC, and precipitation. Their intercorrelation was of an acceptable level for the analysis (Sect. 2.4). Daily FCOS was best explained by VPD, $R_n$, $T_s$, and SWC ($R^2 = 0.65$, Table 1), where the contribution of SWC was minor (see also Kooijmans et al., 2019). Primary variables directly interacting with the canopy (VPD, PAR, and $T_a$) explained FCOS with $R^2 = 0.53$. On a monthly scale, replacing VPD by precipitation

gave the highest $R^2$ (0.88), while $R^2$ (VPD, PAR, $T_a$) = 0.77. Univariate analysis for single factors revealed that the temperature was the most important factor governing FCOS (Table 1). Maignan et al., (2021) studied the importance of different drivers (PAR, $T_a$, VPD, LAI, SWC) to stomatal conductance in Hyytiälä using random forest models. The three most important drivers for stomatal conductance were (in order) PAR, $T_a$ and VPD. This is well in line with our univariate analysis that ranked temperature and radiation as the most important drivers of FCOS, that is mostly regulated by stomatal

conductance. The importance of VPD is larger on longer time-scales. While some of the interactions are non-linear, as seen from Fig. 3 and Eqs. (8-11), the linear regression analysis still provides information on the relative importance of the environmental variables, as non-linear correlations usually have a high linear correlation as well.

We also applied wavelet analysis (Torrence & Gilbert, 1998). It revealed coherence between FCOS and PAR on daily and

yearly temporal scales without any significant time lags between them (Fig. S6), indicating a rapid response of stomata to PAR. The pattern of the coherence between FCOS and VPD was very similar, although the coherence values were somewhat

smaller than those with PAR. The coherence between $T_a$ and FCOS had a 3-hour lag on a daily scale, so minimum FCOS (uptake maximum) was reached before the $T_a$ maximum. $T_a$ reached its daily maximum usually in the afternoon between 3:00 and 5:00 pm (not shown), while FCOS minima were most frequently between 10:00 am and 2:00 pm (Fig. 2c). This is caused by VPD and PAR, which control the stomatal conductance (Fig. S6, Kooijmans et al., 2019). In addition, Maignan et al., (2021) showed that the internal conductance is driven by $T_a$ and limits the total conductance, especially in the afternoon. On a yearly scale there was no significant phase difference between $T_a$ and FCOS.

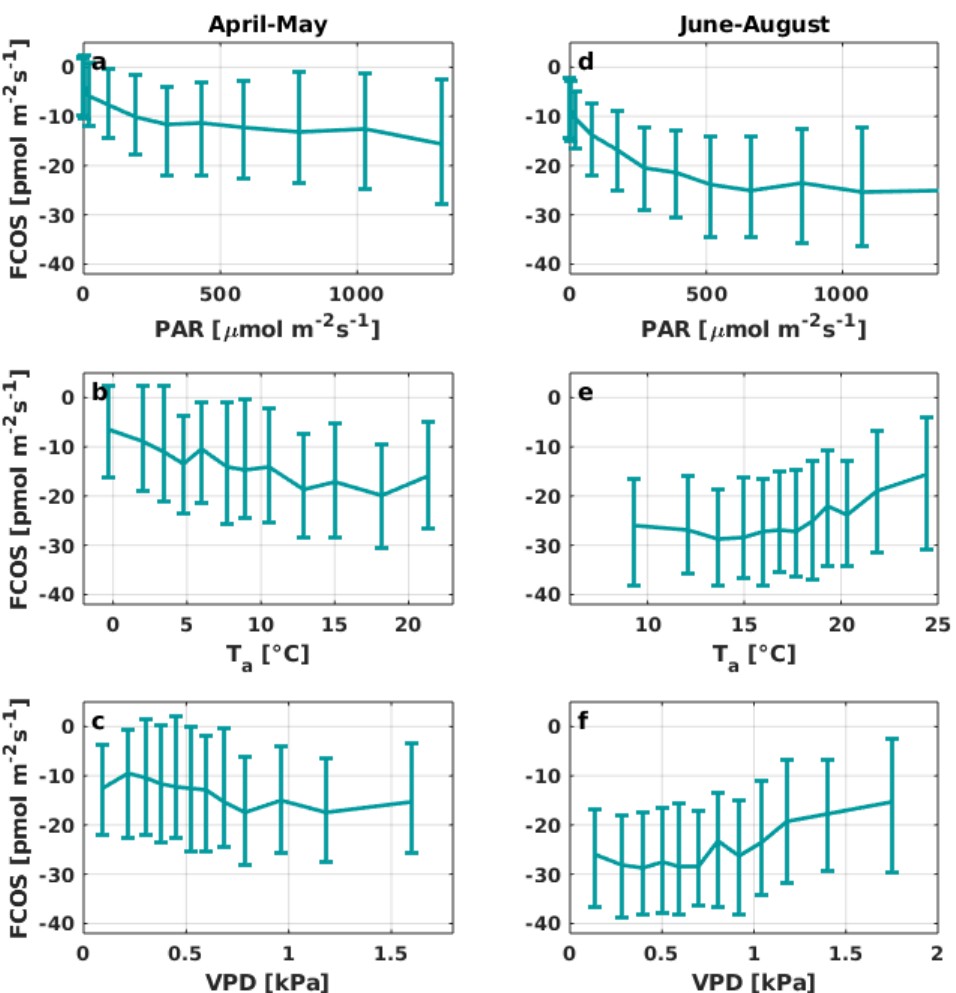

**Figure 3. Carbonyl sulfide flux (FCOS) relationship with environmental variables in spring (a–c) and summer (d–f). Plots b, c, e, and f are filtered to high radiation only (photosynthetically active radiation, PAR > 500 µmol m−2 s−1). Data are binned in 12 equally sized bins and whiskers represent the 25th and 75th percentiles. VPD = vapor pressure deficit.**

| | | Daily | | Weekly | | Monthly | |
|---|---|---|---|---|---|---|---|
| | | $R^2$ | $P$ | $R^2$ | $P$ | $R^2$ | $P$ |
| Multivariate regressions | $VPD + T_s + R_n + SWC$ | 0.65 | <0.001 | 0.86 | <0.001 | — | — |
| | $VPD + T_a + R_n + SWC$ | 0.61 | <0.001 | 0.80 | <0.001 | — | — |
| | $VPD + PAR + SWC + T_s$ | 0.60 | <0.001 | 0.82 | <0.001 | — | — |
| | $VPD + T_a + R_n$ | 0.58 | <0.001 | 0.76 | <0.001 | — | — |
| | $VPD + PAR + T_s$ | 0.55 | <0.001 | 0.79 | <0.001 | — | — |
| | $VPD + T_a + PAR + SWC$ | 0.54 | <0.001 | 0.73 | <0.001 | — | — |
| | $VPD + T_a + PAR$ | 0.53 | <0.001 | 0.73 | <0.001 | 0.77 | <0.001 |
| | $Precip + T_s + SWC + R_n$ | — | — | — | — | 0.88 | <0.001 |
| | $Precip + T_s + SWC + PAR$ | — | — | — | — | 0.86 | <0.001 |
| | $Precip + T_s + SWC$ | — | — | — | — | 0.85 | <0.001 |
| | $Precip + T_a + SWC$ | — | — | — | — | 0.82 | <0.001 |
| Univariate regressions | $T_s$ | 0.46 | <0.001 | 0.68 | <0.001 | 0.71 | <0.001 |
| | $T_a$ | 0.39 | <0.001 | 0.63 | <0.001 | 0.70 | <0.001 |
| | $R_n$ | 0.27 | <0.001 | 0.42 | <0.001 | 0.28 | 0.04 |
| | PAR | 0.13 | <0.001 | 0.28 | <0.001 | 0.24 | 0.03 |
| | $\chi_{COS}$ | 0.13 | <0.001 | 0.19 | <0.001 | 0.24 | 0.03 |
| | VPD | 0.04 | <0.001 | 0.15 | <0.001 | 0.17 | 0.08 |
| | SWC | 0.04 | <0.001 | 0.06 | 0.03 | 0.02 | 0.51 |
| | RH | 0.00 | 0.43 | 0.00 | 0.98 | 0.02 | 0.54 |
| | Precip | 0.00 | 0.24 | 0.12 | 0.002 | 0.49 | <0.00 |

**Table 1. Results of the multivariate and univariate regression analysis for carbonyl sulfide (COS) flux at daily, weekly, and monthly timescales. Variables tested are vapor pressure deficit (VPD), air temperature ($T_a$), photosynthetically active radiation (PAR), net radiation ($R_n$), soil water content (SWC), soil temperature ($T_s$), precipitation (Precip), relative humidity (RH), and atmospheric COS mixing ratio ($\chi_{COS}$). Two variables related to the similar physical quantity were never used in the same model ($T_a$ and $T_s$, PAR and $R_n$, or RH and VPD). All the models which included more than one variable were tested with the variation inflation factor, which was in all cases less than 5, showing that intercorrelation of the variables is negligible.**

## 3.4 Parameterization and simulation of the net carbonyl sulfide flux

Fig. 4 presents the modulation of the four variables of the FCOS, where FPAR and FS have the biggest effect via PAR and $T_a$ governing seasonality and LAI the smallest one. For the parameterization function (Eqs. 8-11) dependencies on their drivers, see Fig. S7. At the same time, PAR varied between 0 and 1870 µmol m$^{-2}$ s$^{-1}$ , $T_a$ between −28 and +29 °C, VPD between 0 and 2.7 kPa, and LAI between 5.3 and 7.1 m$^2$ m$^{-2}$. Eq. 7 provides a robust description of the average FCOS dynamics from a yearly to daily scale ($R^2$ = 0.57) (Fig. 2a, Fig. 5 and Figs S8 and S9, respectively). The difference of the

parameterization to the gap-filling function presented in Kohonen et al., (2020) is that unlike the gap-filling function, this simple parameterization does not require evaluation of parameter values against measurements every two weeks. The evolution of FCOS is taken into account solely with environmental variables. The prediction from Eq. 7 differs from the monthly medians, with at least 20 % of data measured (filled circles in Fig. 2a), mostly in April-June 2013, July-August 2014 and September 2015. In 2016–2017, the predicted values closely follow the measured data excluding April 2016. In April -June 2013 neither VPD nor SWC were significantly higher or lower, respectively, than in other years (Fig. S1). In July 2014 VPD was about twice as high as in 2015-2017, and so FCOS was likely low due to stomatal regulation. In April 2015 SWC is lower than in other years which may similarly explain the low COS flux in that period. In April 2016 neither VPD nor SWC were significantly higher or lower, respectively, than in other years. High VPD or low SWC may also be reflected in the net $CO_2$ eocsystem exchange (NEE). However, NEE values do not differ from other years during these periods of low COS uptake, excluding the highest carbon uptake (lowest NEE), in contradiction to FCOS, in July 2014 (Fig. S1). However, PAR was higher in July 2014 than in any other year, which may explain high carbon uptake. FCOS is more sensitive to the stomatal conductance than the $CO_2$ exchange is, because the changes of the sub-stomatal $CO_2$ concentration are partly buffering the direct effect of the stomatal closure.

The local parameterization and the parameterization with SiB4 data are close to each other, while the SiB4 simulation underestimates FCOS especially during summertime and fall (Fig. 5). Moreover, the decrease in FCOS that was observed in July-August 2014 due to warm and dry conditions was not simulated by SiB4. Instead, SiB4 simulated larger FCOS than other years (Fig. 5, 6) as a result of higher temperatures. It is likely that SiB4 simulates a too small response of the stomata to dry conditions for ENF specifically. A similar result was found for $CO_2$ fluxes by Smith et al. (2020), a study on the European drought in the summer of 2018. They demonstrated that SiB4 does show a drought response, but especially site observations at ENF ecosystems showed a stronger decline in carbon uptake than SiB4 did. In SiB4, ENF is specifically set to be resilient to droughts by setting a lower limit to soil moisture stress on photosynthesis, which is used to derive the stomatal conductance and thereby connects to COS leaf uptake. The COS flux timeseries provide additional evidence that the lower bound on soil moisture stress in ENF ecosystems should be removed. Regarding the general underestimation of FCOS by SiB4, Kooijmans et al. (2021) also found that underestimations of FCOS at Hyytiälä were consistent with underestimations of GPP estimates. Possible methods to increase these fluxes in SiB4 is to increase the maximum carboxylation rate of RuBisCO that is also used to simulate the carbonic anhydrase activity, relevant for COS uptake. Another approach is to incorporate a $CO_2$ fertilization effect that would increase the aboveground biomass and would increase both fluxes. Research is ongoing to implement an accurate representation of the $CO_2$ fertilization effect in SiB4, at the same time assessing other processes like respiration and water use efficiency. Simulations of COS soil uptake in Hyytiälä are also too low (Kooijmans et al., 2021), which could be improved with more accurate carbonic anhydrase uptake parameters specific to the ENF soil. Several studies also suggested the role of bryophytes (Gimeno et al., 2017) and epiphytes (Rastogi et al., 2018), which may play a role in boreal forests, but which are not specifically included in SiB4.

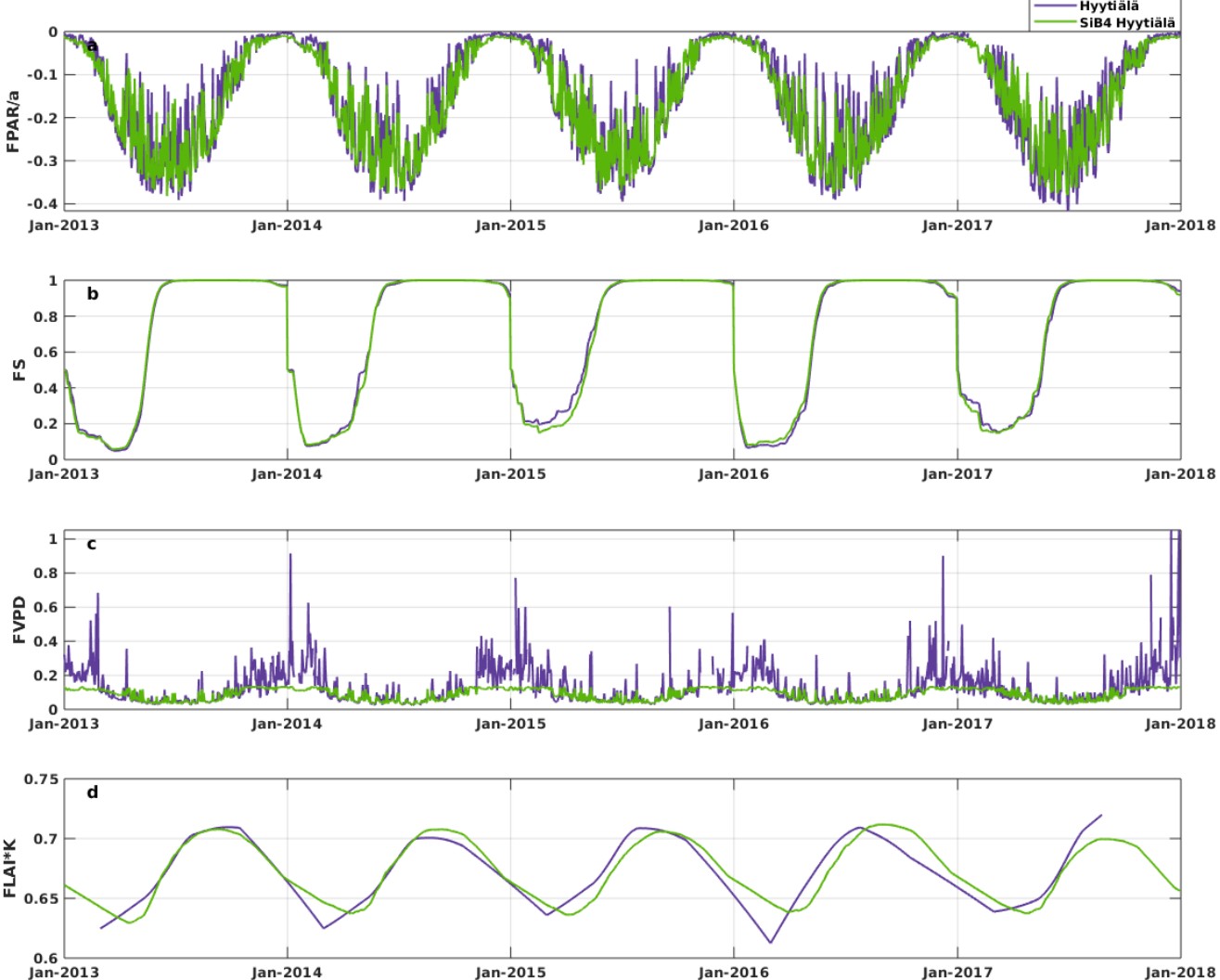

**Figure 4. Time series of the daily average values of the parameter functions (Eqs 2–5) normalized to vary between 0 and 1. The purple line represents the parameterization and green line the parameterization with SiB4 in Hyytiälä. FLAI = foliage and canopy light penetration, FPAR = stomatal response to PAR, FS = phenology of biochemical reactions, FVPD = stomatal regulation.**

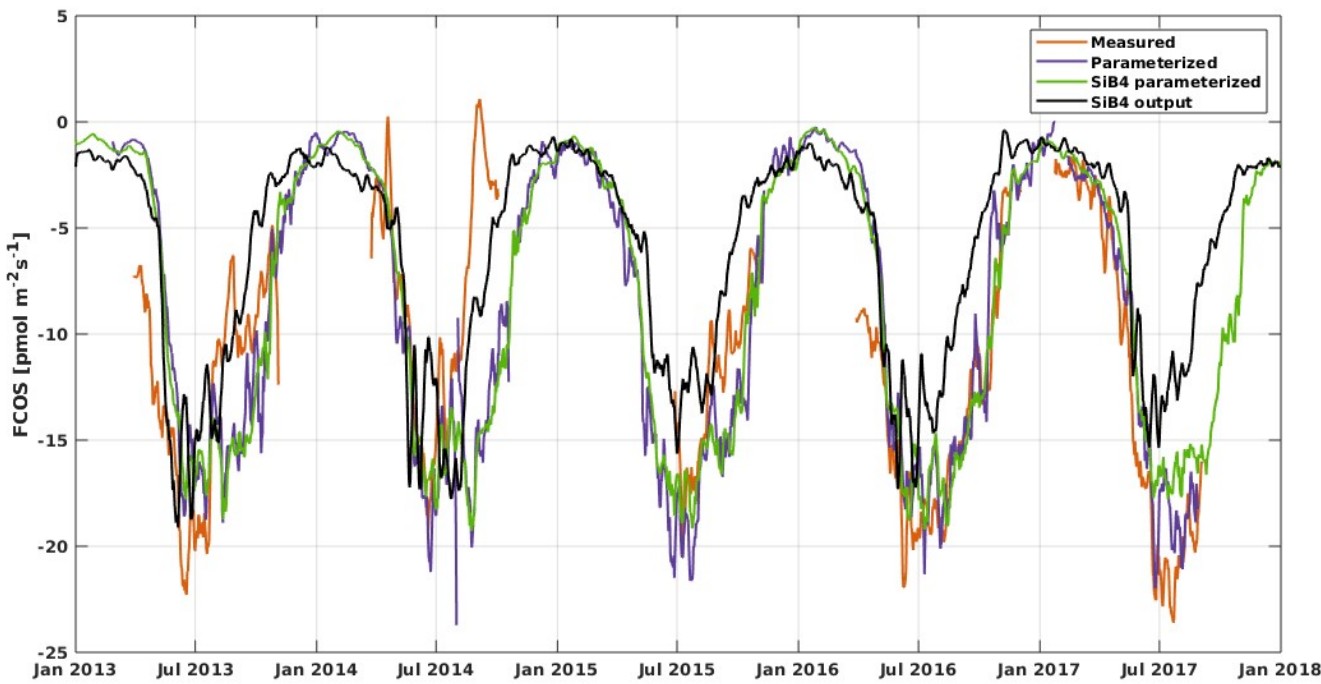

**Figure 5. Weekly averaged FCOS from measurements (gap-filled, orange line), parameterization (purple line), SiB4**
**parameterization (green line) and standard SiB4 output (black line) in Hyytiälä.**

### 3.5 Carbonyl sulfide balances and their interannual variation

We analyzed COS balances, the share between day- and nighttime uptake, and their interannual variability. The cumulative
FCOS (FCOS$_{cum}$) values are presented in Fig. 6 for the periods of the smallest amount of gap-filled data, that is, July–August
in 2013–2017. FCOS$_{cum}$ in 2014 was 52 % lower than in 2017. The larger total uptakes were not only due to daytime uptake.
The higher absolute nighttime uptake corresponded not only to the higher total uptake but also to the higher nocturnal
percentage of the total uptake. The cumulative nocturnal uptake fraction of the total uptake was 0.14, 0.12, 0.15, 0.17, and
0.21 for years 2013–2017, respectively. The total FCOS$_{cum}$ for the period April–August was $-183 \pm 41$, $-130 \pm 30$,
$-207 \pm 45$, $-205 \pm 45$ µmol COS m$^{-2}$ for years 2013, 2014, 2016, and 2017, respectively. Expressed as sulfur uptake the
FCOS$_{cum}$ becomes $-58.5 \pm 13$, $-41.6 \pm 9.6$, $-66.4 \pm 14.4$, $-65.5 \pm 14.4$ gS ha$^{-1}$ for years 2013, 2014, 2016, and 2017,
respectively. We can compare the amount of sulfur deposited by COS with the dominant sulfur compounds. The annual
sulfur deposition of sulfate and sulfur dioxide (SO$_2$) is estimated to be 780 gS ha$^{-1}$ (pers. comm. Hannele Hakola, Finnish
Meteorological Institute) for the measurement site. The COS sulfur deposition reported here was thus 7 % of that of sulfate
and SO$_2$. The parameterized FCOS$_{cum}$ (both using local meteodata and SiB4 meteodata) shows higher total uptake than
FCOS measurements for years 2013-2015 but lower uptake for years 2016 and 2017 (Fig. 6). While both the measured and

parameterized FCOS$_{cum}$ have a slight decreasing trend during 2013-2017 (uptake increasing), the SiB4 output shows an opposite trend of increasing FCOS$_{cum}$ (decreasing total COS uptake). However, this time series is too short analysing trends and the observed differences can be explained by differences in environmental drivers.


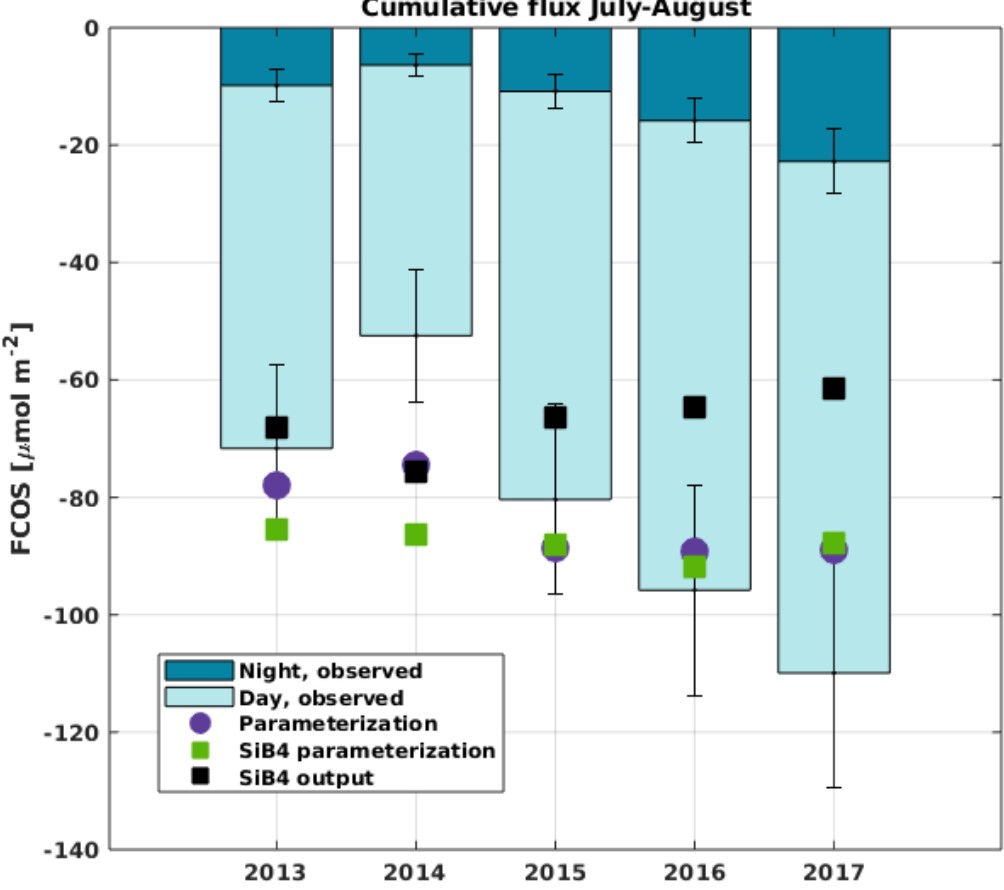

**Figure 6. Cumulative sum of gap-filled carbonyl sulfide flux (FCOS) in 2013–2017 during the period July–August, separated into day and night contributions. The percentage of gap-filled COS data during nighttime and daytime, respectively, was 77 and 58 % in 2013, 70 and 45 % in 2014, 57 and 38 % in 2015, 65 and 43 % in 2016, and 55 and 29 % in 2017. Purple circles represent the parameterized FCOS, green squares the parameterization by SiB4 and black squares the standard SiB4 output for Hyytiälä.**

**3.6 Implications for global biogeochemical cycles**

The long-term time series presented in this study, together with the relationships of the FCOS with PAR, $T_a$, VPD, and LAI, can help to evaluate and improve biosphere models and thereby contribute to an accurate biosphere sink estimate. We calibrated the SiB4 PAR, LAI, VPD and $T_a$ for the grid cell where Hyytiälä is located against observations (see Fig. S10). We then applied the FCOS parameterization to the calibrated SiB4 meteorological and phenological data for the whole

boreal region to evaluate FCOS simulations by SiB4. The resulting daytime average FCOS in the boreal region is always larger than the FCOS simulated by SiB4, especially in the summer months (Figs. 5-7, see Fig. S11 for the difference between the two methods). The total FCOS of evergreen needleleaf forests (ENFs) in the boreal region is estimated to be -14.6 Gg S y$^{-1}$ by the parameterization for the years 2013–2017, which is close to the estimate for boreal forests (19.2-33.6 Gg S y$^{-1}$) by Sandoval-Soto et al., (2005), but 1.5 times larger than that simulated by SiB4 (-10.6 Gg S y$^{-1}$). These results are in line with

the recent top-down studies of the COS atmospheric budget (Ma et al., 2021; Remaud et al., 2021; Hu et al., 2021), who pointed to a missing sink in the higher latitudes of the northern hemisphere. The parameterization presented here could serve to improve the prior descriptions of the COS biosphere flux used for inverse modeling studies like that of Ma et al. (2021). Such an approach would provide more accurate COS biosphere sink estimates and could help to improve the representation of gross primary production in biosphere models as well.

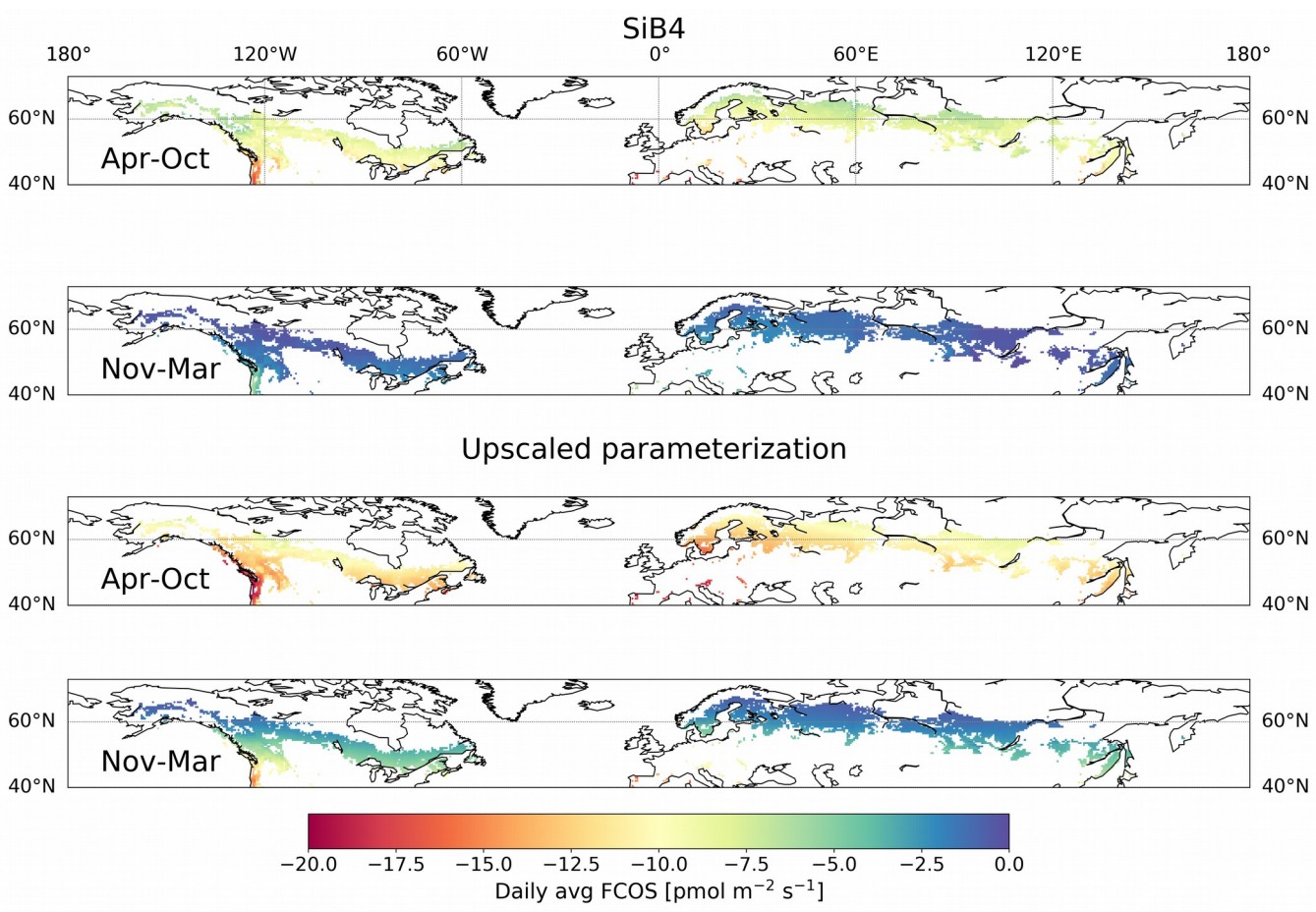

**Figure 7. Daily average (avg) carbonyl sulfide flux (FCOS) in the boreal region based on SiB4 (top) and upscaled parameterization simulations (bottom) for the periods April–October and November–March. The boreal region is defined as the area where evergreen needleleaf forests cover more than 30 % of the land area in the northern hemisphere.**

## 5 Conclusions

In summary, to get a more accurate picture of temporal dynamics and variations of the FCOS between the atmosphere and biosphere, multiyear observations are needed. Here a clear relationship between the spring recovery and the FCOS and temperature thresholds or sums was not observed. The significant reduction of the FCOS under large VPD values on the ecosystem scale found here corroborates the relationship observed between the shoot-scale FCOS and VPD at the same forest by Kooijmans et al. (2019), who illustrated the stomatal limitation of the flux. Wavelet analysis of the ecosystem fluxes confirmed earlier findings from branch-level fluxes at the same site and revealed a 3-hour lag between FCOS and $T_a$ in the daily scale, while no lag between PAR and FCOS was found. The contribution of COS to the total annual sulfur deposition was estimated to be 7 %. We hypothesized that the long-term FCOS can be described by a simple semi-empirical parameterization that employs PAR, $T_a$, VPD, and LAI. We tested the hypothesis with our long time series. We proved the hypothesis by presenting the first easy-to-use parameterization for the FCOS. We scaled up the FCOS parameterization to the whole boreal region and the obtained results are in line with the inverse modeling study by Ma et al. (2021), who pointed to a missing sink in the higher latitudes of the northern hemisphere. We provide recommendations for model development that could resolve this gap. It remains to be studied whether multiyear dynamics and seasonal patterns in the in situ flux analyzed here are reflected also in regional terrestrial COS air concentration and sink observations and estimates.

## Acknowledgments

Special thanks to Helmi Keskinen, Sirpa Rantanen, Janne Levula, and other Hyytiälä technical staff for all their support with the measurements. The authors thank the Academy of Finland Centre of Excellence (118780), Academy Professor projects (312571 and 282842), ICOS Finland (3119871), ACCC Flagship funded by the Academy of Finland grant number 337549 and the Tyumen region government in accordance with the Program of the World-Class West Siberian Interregional Scientific and Educational Center (National Project "Nauka"). K.-M. Kohonen thanks the Vilho, Yrjö and Kalle Väisälä foundation for its support. L.M.J. Kooijmans thanks the ERC advanced funding scheme (COS-OCS, 742798). We acknowledge the computing resources from the Netherlands Organization for Scientific Research (NWO; grant no. NWO-2021.010).

## Author contributions

TV, IM and KMK designed the study. KMK, PK and AP performed the measurements and flux processing. MZ and DN helped with the flux measurements and setup. KMK, LMJK and LF performed the data analysis: KMK developed the local flux parameterization and LMJK expanded the parameterization for the boreal region, while LF did the multivariate analysis. MK, JB and DY commented the manuscript. TV and KMK wrote the manuscript with contributions from all co-authors.

**Competing interests**

The authors declare that they have no conflict of interest.

**Data availability**

The flux data and SiB4 simulated COS fluxes used in this study are available at https://doi.org/10.5281/zenodo.5906705. Environmental data used in the study are available in the AVAA – Open research data publishing platform (https://smear.avaa.csc.fi/). The metadata of the observations are available via the Etsin service.

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
