# Peer review of "Long-term fluxes of carbonyl sulfide and their seasonality and interannual variability in a boreal forest"

_Atmospheric Chemistry and Physics, 2021_

## Referee Comment (RC3)

**Review : Long-term fluxes of carbonyl sulfide and their seasonality and interannual variability in a boreal forest**

**1 GENERAL COMMENTS**

This paper analyses the seasonal and interannual variabilities of 5-years biospheric COS fluxes at a site located in a boreal pine forest in Finland. To explain these variability modes, the relationships between the COS biospheric sink and environmental drivers (vapor pressure deficit, light, air temperature) are described. A linear regression model is used to select the main environmental drivers of COS biospheric sink variability. They further develop an empirical model of the COS biospheric sink that is function of PAR, LAI and VPD. The model calibrated with the observations successfully reproduces the observed seasonal and interannual variabilities. Then, they optimize the parameters of the model using the LAI, VPD, PAR simulated by the SIB4 Land Surface Model along with the observed COS fluxes at the site of interest. The calibrated model is then applied to the evergreen boreal needleleaf forests over the whole northern hemisphere. The total COS biospheric sink is greater than the simulated one by the SIB4 LSM, in agreement with a missing sink inferred by most top-down studies.

Overall, the paper provides a valuable advance in reconciling the bottom-up and top-down COS budget at high latitude. However, the manuscript seems to be written in a rush. The many references in the Supplementary makes it difficult to follow sometimes and some clarifications are needed. In particular, I have a few questions that need to be addressed.

1. In agreement with reviewer 2, the empirical relations 2-5 call for more explanations of their provenance and their physical meanings in the main text. In the equation 5, it is unclear how the parameter e is fixed. For the observations, how and why was the parameter e fixed before optimizing the other parameters (Page 4, line 107)? For the SIB4 LSM, why is e multiplied by 2.1, the average ratio of Hyytiala and SIB4 LAI data (Page 5 line 2)?

2. This concerns the text around Table 1, where the authors attempt to disentangle the drivers of the COS biospheric sink variability. Given the high non-linearity of the underlying equations 2-5, it is not clear how well the simple multiple linear regression analysis presented in Table 1 is able to capture such highly non-linear interactions. To my opinion, the statistical analysis would require a different approach, such as a decision tree or a neural network in order to deal with the highly non-linear interactions between variables.

3. To upscale the flux observations to evergreen needleleaf forests in the whole boreal region, it would have been more straightforward to use the MODIS products of PAR, LAI along with the surface air temperature, VPD from the MERRA reanalysis. The MODIS-derived LAI and PAR should be more realistic spatial variability than the ones simulated by the LSM. A processed-based alternative would have been to (i) calibrate the parameters of the SIB4 LSM with the

observed LAI, PAR, VPD, FCOS at the closest grid point of the eddy covariance site and (ii) to run the newly calibrated LSM to simulate the COS biospheric fluxes over evergreen needleleaf forests in the whole boreal region. Such analysis would have shed light on the specific LSM parameters that need to be calibrated. The use of the SIB4 LSM would be better justified if the authors evaluate the performance of the LSM to reproduce the COS biospheric sink at Hyytiala (for instance Fig.4). Also, the statistical analysis in Table 1 could be also done for the SIB4 LSM and compared with Maignan et al., 2021 who did a similar analysis.

**2  SPECIFIC COMMENTS**

Page 2, line 41-42: "The terrestrial ... 1360 GgS/y" I would add Remaud et al. (2021) who used a more recent anthropogenic inventory to infer the net biospheric sink over the globe through inverse modelling. The missing sink in the high northern latitudes was also shown in Remaud et al. (2021); Hu et al. (2021).

Page 3, line 7: "of which is gap-filled ..." The method for gap-filling needs to be explained for the sake of clarity.

Page 3, line 91: I would add "at 23 m height, where the EC measurements were made".

Page 3, line 95: Where are located the 5 locations?

Page 4, line 106: Here, I would present the parameterization of Carbonyl sulfide fluxes by showing and explaining the equations 1-5.

Page 6, line 146, "However, the snow ... region." What is the link with the former sentence?

Page 6, line 153: "We analyzed ... FCOS" What is the dependence of the CO2 flux to the VPD?

Page 7, line 165: Epiphytes can also significantly take up the atmospheric COS during the night when the soil is wet as shown by Rastogi et al. (2018).

Page 7, line 182: What does "unfiltered" data mean? Was the Fig. S6 drawn with the data that have not been gap-filled?

Page 7, line 201: The set-up of the wavelet analysis should be explained in the method. Figure S7 should be in the main text.

Page 7, lines 194-209: See Maignan et al. (2021) (Fig. 3). Based on the ORCHIDEE LSM, they showed that the internal conductance was mainly driven by Ta. In the afternoon, the internal conductance limits the total conductance and reaches a maximum 3 hours after the peak in stomatal conductance.

Page 9, Table 1: I wonder how the result would look like if the measurements were not gap-filled.

Page 10, equations 1-5 : A plot associated with each equation (e.g FPAR as function of PAR) would give a better idea of their contribution to FCOS.

Page 10, line 247: It should be mentioned earlier in Part 2.4 that the parameterization is not applied to the gap-filled data.

Page 10, line 252: It is worth discussing here the reasons why the empirical model and the observations for July 2014 are in disagreement for July 2014 (Figure 1). Does the empirical model underestimate the sensitivity of the surface fluxes to vapor pressure deficit? Does the net CO2 fluxes exhibit such decrease during the same period?

Page 11, line 263: "The cum.. respectively." What are the numbers within parenthesis? IS FCOScum computed using the gap-filled measurements? Page 11, lines 265-267: I don't understand the link between the COS sulfur deposition and the Carbonyl

sulfide balances and their interannual variation. Please explain.

Page 12, Figure 4: I find Figure 4 particularly interesting, showing that the COS biospheric sink increases between 2013-2017 while the average atmospheric COS concentration decreases over the whole northern hemisphere during the same period. Has the length of the growing season been increasing over the years? It would be interesting to investigate the reasons underlying

5    this increase, for instance by looking at the environmental drivers of FCOS. It would also be interesting to reproduce the Figure 4 for the SIB4 LSM.

Page 12, line 273: "..improve the .. sink estimate." and line 285 "... could help to improve the representation of gross primary production in biospheric models as well.". The paper falls a bit short of really providing a parameterization that could help to improve the LSMs. This is more true that this study highlights a LSM bias which is an underestimation of the simulated

10   biospheric sink in the high latitudes, in agreement with recent inverse modelling studies (Ma et al., 2021; Remaud et al., 2021). Page 12, line 273: I would rather say: "with the recent top-down studies of the COS atmospheric budget (Ma et al., 2021; Remaud et al., 2021; Hu et al., 2021)"

**References**

Hu, L., Montzka, S. A., Kaushik, A., Andrews, A. E., Sweeney, C., Miller, J., Baker, I. T., Denning, S., Campbell, E., Shiga, Y. P., Tans, P., Siso, M. C., Crotwell, M., McKain, K., Thoning, K., Hall, B., Vimont, I., Elkins, J. W., Whelan, M. E., and Suntharalingam, P.: COS-derived GPP relationships with temperature and light help explain high-latitude atmospheric $CO_2$ seasonal cycle amplification, Proceedings of the National Academy of Sciences, 118, https://doi.org/10.1073/pnas.2103423118, https://www.pnas.org/content/118/33/e2103423118, publisher: National Academy of Sciences Section: Physical Sciences, 2021.

Ma, J., Kooijmans, L. M. J., Cho, A., Montzka, S. A., Glatthor, N., Worden, J. R., Kuai, L., Atlas, E. L., and Krol, M. C.: Inverse modelling of carbonyl sulfide: implementation, evaluation and implications for the global budget, Atmospheric Chemistry and Physics, 21, 3507–3529, https://doi.org/10.5194/acp-21-3507-2021, https://acp.copernicus.org/articles/21/3507/2021/, publisher: Copernicus GmbH, 2021.

Maignan, F., Abadie, C., Remaud, M., Kooijmans, L. M. J., Kohonen, K.-M., Commane, R., Wehr, R., Campbell, J. E., Belviso, S., Montzka, S. A., Raoult, N., Seibt, U., Shiga, Y. P., Vuichard, N., Whelan, M. E., and Peylin, P.: Carbonyl sulfide: comparing a mechanistic representation of the vegetation uptake in a land surface model and the leaf relative uptake approach, Biogeosciences, 18, 2917–2955, https://doi.org/10.5194/bg-18-2917-2021, https://bg.copernicus.org/articles/18/2917/2021/, publisher: Copernicus GmbH, 2021.

Rastogi, B., Berkelhammer, M., Wharton, S., Whelan, M. E., Itter, M. S., Leen, J. B., Gupta, M. X., Noone, D., and Still, C. J.: Large Uptake of Atmospheric OCS Observed at a Moist Old Growth Forest: Controls and Implications for Carbon Cycle Applications, Journal of Geophysical Research: Biogeosciences, 123, 3424–3438, https://doi.org/https://doi.org/10.1029/2018JG004430, https://agupubs.onlinelibrary.wiley.com/doi/abs/10.1029/2018JG004430, 2018.

Remaud, M., Chevallier, F., Maignan, F., Belviso, S., Berchet, A., Parouffe, A., Abadie, C., Bacour, C., Lennartz, S., and Peylin, P.: Plant gross primary production, plant respiration and carbonyl sulfide emissions over the globe inferred by atmospheric inverse modelling, Atmospheric Chemistry and Physics Discussions, pp. 1–43, https://doi.org/10.5194/acp-2021-326, https://acp.copernicus.org/preprints/acp-2021-326/, publisher: Copernicus GmbH, 2021.

---

## Author Response (AR1)

Reviewer comments in black
Author response in blue
*Text in the revised manuscript in italic*

**Reviewer #1** - Jürgen Kesselmeier
General View
I was glad to read this report about long-term series of measurements of carbonyl sulfide fluxes over boreal forests. This 5-years study delivers a large data set and a great overview about the seasonality and interannual variability. Furthermore, it greatly supports the process study-based knowledge that stomatal conductance is one of the keys to interpret the flux behavior under day and night conditions, though the final consumption of COS is light independent, based on the enzymatic degradation by several enzymes. The results of this strong project must be published. However, the current version can and should be improved. The weaknesses of the current parameterization should be discussed. This parameterization is only a first attempt to interpret the monthly fluxes and embed the measurement data into a wider description based on environmental factors, such as the measured photosynthetically active radiation, vapor pressure deficit, air temperature, and leaf area index. The procedure is valid to get an overview but this parameterization obviously still needs some fitting parameters. What does this fitting mean? Altogether, the fitted simulation matches the general exchange pattern over several seasons in a satisfactory manner. However, the total net flux is a result of a complex process which is affected by contributions of trees, cryptogamic covers and soils which may contribute significantly as it has been demonstrated earlier by several process-based studies (see review by Whelan et al. 2018). Here a complete overview about the environmental factors will be of help for the reader to understand.

We thank the reviewer for the very insightful and constructive comments. Related to parameterization and its physical meaning, we wrote in the previous manuscript version: "*FPAR describes the stomatal response to PAR, FS the phenology of biochemical reactions, FVPD the stomatal regulation, and FLAI the amount of foliage and canopy light penetration.*" The parameter functions are related to CO2 exchange and stomatal regulation from earlier studies (Mäkelä et al., 2008; Dewar et al., 2018, Peltoniemi et al., 2015), but modified and fitted to our COS flux data. We focus on ecosystem net fluxes without partitioning them into components such as canopy exchange or soil exchange. We understand that the parameterization mainly takes into account the canopy processes, but as we have seen from soil flux measurements at Hyytiälä (Sun et al. 2018), the COS soil flux does not have a diurnal pattern and changes only slightly throughout the season (ca. from -3 to -4 pmol m$^{-2}$s$^{-1}$), compared to a high seasonal variation in the net ecosystem COS exchange (ca. from -2 to -22 pmol m$^{-2}$s$^{-1}$). This variation is taken into account in the variation of the parameterization function FS, that takes into account the seasonal variation in phenology. PAR, T$_a$ and VPD were shown in Supplement Fig. S2, LAI is now also added in the same figure in the new version of the manuscript and the figure was moved to the main text (Fig. 1). We have also added a new section 3.1 "Environmental conditions" to the Results section, describing the meteorological conditions of the measurement period, as suggested. The weaknesses of the parameterization are addressed later in the response, in the specific requests.

Specific Requests
One of the most important products of this paper is the development of a parametrization for FCOS. The reader automatically searches for information under Material and Methods. However, there one finds only a short chapter about the fitting parameters. I propose to shift the description with the corresponding formulas from the Results-section to Material and Methods. Furthermore, the information that fitting parameter were retrieved with MATLAB is not really satisfying. I miss information about the basics, i.e. PAR, Ta, VPD, and total LAI data over the 32 months. This information is needed to get a feeling for interannual and seasonal fluctuations. Why not producing a figure giving an overview about these environmental keys in a similar manner as for the fluxes in

figure 1? Thus, the reader would get a picture about the seasonal microclimate. Eddy correlation measurements deliver data for fluxes under turbulent conditions. I miss some critical remarks concerning nighttime flux data. As discussed by the authors, higher vegetation with a well-adapted gas exchange under stomatal regulation can be ruled out, though not completely. However, there can be a strong uptake by soils and cryptogams during the night which becomes strongly visible under a stable nocturnal boundary layer. The authors interpret non-zero uptake value in the dark simply as the ecosystem nocturnal COS uptake. Could this be specified and related to soil and cryptogams? The finding that the temperature is the most important factor governing FCOS at least gives room to have a closer look.

We have moved the parameterization explanation from Results to Material and Methods section, as well as all methodological explanations from the Supplementary material have now been moved to the main text. We have clarified the fitting parameter optimization in the text as follows: *"Parameters a = −341.81, b = 1000, c = −0.77, and d = 1.03 were optimized using MATLAB's fminsearchbnd function to find the smallest root mean square error of the parameterized FCOS against the measured (non-gap-filled) FCOS. The fminsearchbnd function finds the minimum of a constrained multivariable function using a derivative-free method (D'Errico, 2021). Parameters were given upper and lower limits of a ϵ [5,500], b ϵ [10,1000], c ϵ [-3,3], d ϵ [1,5]. It is to be noted that parameter b is at its upper limit. However, increasing the limit (until infinity) for parameter b only resulted in a higher value for parameter a, without significantly improving the overall fit. Parameter e = 0.18 was fixed before optimizing the other parameters according to a previous study by Peltoniemi et al., (2015), since parameter e is related to ecosystem phenology specific to the site."*

We have moved the texts describing eddy covariance measurements, multivariate analysis, wavelet coherence analysis and environmental variables from the supplementary material into the main text in the Materials and Methods section to make the text easier to follow for the reader.

PAR, $T_a$ and VPD were shown in Supplement Fig. S2 but now moved to the main text as Fig. 1 and LAI is now also added in the same figure in the new version of the manuscript. We have also added description about the meteorological conditions and anomalies of the environmental variables to the Results section. Monthly medians of the environmental parameters are shown in Supplement Fig. S1.

EC flux data from nighttime with low turbulence conditions have been filtered out using the standard u* filtering, so that all data with u*<0.3 m/s have been filtered out, as explained in the Materials and Methods section. The soils act as a COS sink in Hyytiälä during nighttime, as shown by Sun et al., (2018). The nighttime soil flux during summer is ca. -3 pmol $m^{-2}s^{-1}$, which could partly explain the net ecosystem nighttime COS sink (ca. -10 pmol $m^{-2}s^{-1}$), although not fully and we do see the stomata staying open during night (Kooijmans et al. 2019). We have added a few sentences about the importance of soil fluxes during nighttime: *"Mosses can also significantly take up COS during the night when the soil is wet (Rastogi et al., 2018). Nighttime soil uptake in Hyytiälä was ca. -3 pmol $m^{-2}s^{-1}$ (Sun et al., 2018) while the ecosystem scale nighttime uptake in our study is ca. -10 pmol $m^{-2}s^{-1}$. However, we did not measure the contribution from mosses. The nighttime COS uptake in Hyytiälä is thus likely a combination of soil and moss uptake, but also has a larger contribution from the canopy (Kooijmans et al., 2017)."*

The simple semi-empirical parametrization, as the authors call it, is very helpful. However, I miss a holistic overview based on this promising long-term study at Hyytiälä. Which information can be given about the meteorological background, plant development and seasonal behavior? Where are reports about simultaneous measurements of exchanges with canopy and soils and the atmosphere? I expect this diversity is of special importance to handle flux measurements above a boreal forest which can present a quite open structure.

Meteorological background ($T_a$, PAR, VPD, $T_s$, SWC, heat sum) were given in supplement Figs. S2-S5, but we have decided to move one of them to the main text as Fig. 1 and added LAI timeseries to the plot. Additional Figures S1, S3 and S4 in the Supplemet provide information on soil T, SWC, heat sum and monthly medians. The S parameter (and LAI) describe the plant development as well as seasonality. This study focuses on ecosystem scale net fluxes and interactions, so canopy and soil fluxes are not provided separately. We do have soil and branch fluxes measured from few campaigns (Sun et al., 2018; Kooijmans et al., 2019, referenced e.g. in the Introduction: *"For canopy and soil COS exchange in the studied pine forest, see Kooijmans et al. (2017, 2019) and Sun et al. (2018), respectively. "*) but they do not cover the whole 5 years of ecosystem measurements presented in this study and are thus not included. However, many of the branch scale FCOS responses (Kooijmans et al., 2019) were also produced at the site level in this study. We wrote in Section 3.3: *"When $T_a$ was high (Fig. 3e), in theory favoring the uptake due to enhanced biochemical reactions in the mesophyll, the simultaneous occurrence of high VPD (Fig. 3f) limited the uptake due to smaller stomatal conductance values (see Kooijmans et al., 2019). …. Daily FCOS was best explained by VPD, $R_n$, $T_s$, and SWC ($R^2 = 0.65$, Table 1), where the contribution of SWC was minor (see also Kooijmans et al., 2019). …… The coherence between $T_a$ and FCOS had a 3-hour lag on a daily scale, so minimum FCOS (uptake maximum) was reached before the $T_a$ maximum. $T_a$ reached its daily maximum usually in the afternoon between 3:00 and 5:00 pm (not shown), while FCOS minima were most frequently between 10:00 am and 2:00 pm (Fig. 2c). This is caused by VPD and PAR, which control the stomatal conductance (Fig. S6, Kooijmans et al., 2019). In addition, Maignan et al., (2021) showed that the internal conductance is driven by $T_a$ and limits the total conductance, especially in the afternoon."*

There are some disagreements between measured data and simulated ones as indicated in figure 1 for the years 2013 and 2014. This is only poorly addressed in the text. Are there any environmental factors to be made responsible? Meteorology? Plant development? See my comments above. A report about the environmental factors which are the basis for the parameterization will clearly help. Within this context, vapor pressure deficits not only affect higher plants with stomata but also the water content of cryptogamic tissues.

We agree with the reviewer that the disagreement between measured and simulated flux data need to be better addressed in the text. We tried to take the disagreement into account in the parameterization so that only 2014 data was used in determining the fitting parameters. However, this resulted only in a better fit in 2014, but a worse fit for all other years. We found that there is not enough data from periods with high VPD, so that we could do a reliable parameterization. We have added a subplot showing the monthly NEE averages for each year (Fig. S1) because Reviewer #3 wrote: *Does the net CO2 fluxes exhibit such decrease during the same period*?" We have added a paragraph discussing the differences in Section 3.4: *"In April -June 2013 neither VPD nor SWC were significantly higher or lower, respectively, than in other years (Fig. S1). In July 2014 VPD was about twice as high as in 2015-2017, and so FCOS was likely low due to stomatal regulation. In April 2015 SWC is lower than in other years which may similarly explain the low COS flux in that period. In April 2016 neither VPD nor SWC were significantly higher or lower, respectively, than in other years. High VPD or low SWC may also be reflected in the net $CO_2$ eocsystem exchange (NEE). However, NEE values do not differ from other years during these periods of low COS uptake, excluding the highest carbon uptake (lowest NEE), in contradiction to FCOS, in July 2014 (Fig. S1). However, PAR was higher in July 2014 than in any other year, which may explain high carbon uptake. FCOS is more sensitive to the stomatal conductance than the $CO_2$ exchange is, because the changes of the sub-stomatal $CO_2$ concentration are partly buffering the direct effect of the stomatal closure. "*

The light saturation of FCOS (Fig. 2d) is interesting. How is this understood? The consumption of COS by the enzyme carbonic anhydrase is light independent and stomatal restriction under the

given light intensities should not be expected. Besides carbonyl anhydrase, a contribution of the carboxylation enzymes phosphoenolpyruvate carboxylase and ribulose-1.5-bisphosphate carboxylase has been already reported earlier (Protoschill-Krebs and Kesselmeier, Bot. Acta 105, 1992, 206-212) and may help to understand a light saturation. Interesting to see this within flux data. Is this light saturation incorporated into the parameterization?

The light saturation is mainly related to stomatal control, although other factors, such as enzyme contribution, may have a role as well. From eddy covaraince measurements, however, it is impossible to separate between stomatal control and enzyme function processes. Under increasing radiation, the stomatal conductance increases allowing more COS to enter the leaves. Under high radiation, the stomata are as open as they can, causing a saturation. What could limit the COS flux in the leaf level at this point would be reduced humidity in the atmosphere, causing the stomata to close a bit to reduce water loss (Kooijmans et al., 2019). The FPAR function (that includes all light dependent reactions) has a shape like the light saturation, shown in the new Fig. S7. We have included discussion in the text about the role of light-dependent enzymes in COS uptake: *"The fluxes showed a clear relationship with PAR (Figs 3a and d), even though the COS biochemical reactions and CA activity are light independent and FCOS responds to light mainly due to the light response of stomatal conductance (Kooijmans et al., 2019). Besides CA, carboxylating enzymes ribulose-l,5-bisphosphate-carboxylase (RuBisCO) and phosphoenolpyruvate-carboxylase (PEP-Co), that are light dependent, contribute to COS metabolism (Protoschill-Krebs et al., 1992). However, it is not possible to separate between these processes from EC flux measurements and it has been shown that the main enzyme contributing to COS uptake in plants is CA (Protoschill-Krebs et al., 1996). Thus, we do not expect the role of PEP-Co and RuBisCO light-dependency to exceed that of stomatal regulation."*

A remark concerning the numbers derived. Sandoval-Soto et al. (Biogeosciences, 2, 125–132, 2005) made the first attempt to recalculate the global budget for the COS vegetation sink based on GPP but corrected by measured COS/CO2 uptake ratios (experimentally obtained deposition velocities). They used older primary productivity data (Whittaker, R. H. and Likens, G. E.: The biosphere and man, in: The Primary Productivity of the Biosphere, edited by: Lieth, H. and Whittaker, R. H., Springer Verlag, New York, 305–328, 1975) to upscale COS sinks. This attempt demonstrated a clear underestimation of the global vegetational sink at that time and initialized re-estimations. Within their data sets, they came to a number of 0.036-0.063 Tg [COS] per year for the uptake by boreal forests, which is equivalent to 19.2-33.6 Gg S per year. Interestingly, this number is ranging quite close to the 16.6 Gg as estimated by the authors on the current Hyytiälä data. It is interesting to see the developing numbers in view of all the uncertainties. Within this context, I would like to propose to change a principle of citation. In the first and second line on page 2 there is a number of papers cited to demonstrate the relationship between COS and gross carbon uptake. The choice looks arbitrary. I think it would be adequate better to skip all the citations and cite one recent review paper Whelan et al. (Biogeosciences, 15, 3625–3657, 2018) instead. The history and number of important contributions to this special topic are much larger and the proposed review is giving a more complete story.

It is indeed interesting the FCOS sink estimate for boreal forests by Sandoval-Soto et al., 2005 ranges close to our parameterization, thank you for noting! We have added a note about this in the text: *"The total FCOS of evergreen needleleaf forests (ENFs) in the boreal region is estimated to be -14.6 Gg S y⁻¹ by the parameterization for the years 2013–2017, which is close to the estimate for boreal forests (19.2-33.6 Gg S y⁻¹) by Sandoval-Soto et al., (2005), but 1.5 times larger than that simulated by SiB4 (-10.6 Gg S y⁻¹)."*

We agree with the reviewer that the choice of citations in line 34 was somewhat arbitrary. We changed the citations to *"(e.g., Sandoval-Soto et al., 2005; Asaf et al., 2013; Whelan et al., 2018)"* to represent chamber and eddy covariance measurements of COS as well as the synthesis study.

**Reviewer #2** - Mary Whelan

The eddy flux covariance data from this research effort is a valuable contribution to our community. Vesala and his team have worked hard to advance our understanding of ecosystem carbonyl sulfide (OCS) exchange. It would be appropriate to do more with this data and to set it into a greater context.

The motivation for studying OCS over terrestrial ecosystems is to be able to observe leaf-level behavior, specifically stomatal conductance, at scales larger than a leaf. For over a decade, the great promise of OCS has been to improve the representation of stomatal conductance in land surface models. However, there are other equally important questions to be answered, e.g. the contribution of nighttime stomatal conductance to the atmospheric water budget. Given the potential applications of OCS data, the interpretation and application of this dataset appears incomplete.

We thank the reviewer for the very insightful and constructive comments. It is true that COS flux data have many applications, including stomatal behaviour, water fluxes and GPP. However, as long term flux measurements do not exist form other sites, it is unknown how COS fluxes react to environmental changes. It is thus crucial to study long-term changes and interannual variability of the ecosystem scale COS exchange itself, without tying it to photosynthesis and stomatal conductance. In addition, deriving stomatal conductance is a very difficult task to do from EC measurements and deserves a paper dedicated just to stomatal conductance and water budget. Thus we have restricted this study to focus on COS fluxes alone.

Vesala et al. present an impressive observational dataset. They proceed to optimize a set of equations that comprise a semi-empirical model of OCS (Eq 2-5, the results of the first optimization plotted in Figure 1). In order to compare to the land surface model SiB4, the optimization is performed with the same equations again, but this time using variables from SiB4 for the 0.5 x 0.5 degree grid cell containing the field site. The observations are not on a 0.5 x 0.5 degree scale. SiB4 has OCS leaf and soil exchange coded into it. SiB4 can also be run on the site level. An explantation is needed as to why the optimization was performed twice in this manner. Why not compare the OCS exchange predicted directly by SiB4 to the output of semi-empirical model run with related inputs?

As we are interested in the boreal region COS exchange, we decided to run SiB4 for the Hyytiälä location in the same way as we do for the whole boreal region; that is, on grid level using MERRA input data. It is known that MERRA input typically provides higher PAR than is observed on site as is shown in the Figures below (see also supplementary figure S7 of Kooijmans et al. 2021, biogeosciences,). Based on the review comments we decided it is more clear and straightforward to scale the meteo and phenological data of the SiB4 Hyytiälä grid to match with the in-situ observations and use this scaling for the SiB4 meteo data, allowing to use the same parameters in the COS flux parameterization as in the site-level parameterization. We provide this meteo data scaling in the Supplementary material Fig. S10 and state this in the text *"To obtain COS biosphere fluxes for the whole boreal region based on the FCOS observations in Hyytiälä we calibrated the SiB4 PAR, LAI, VPD and $T_a$ for the grid cell where Hyytiälä is located against observations (see Fig. S10). We then applied the parameterization represented in Eq. 7-11 to the whole boreal region (based on the ENF grid cell selection described in the previous paragraph) using the SiB4 meteorological and phenological data."*. We have added SiB4 parameterization functions to Fig. 4 and the parameterized FCOS (both local and SiB4) as well as SiB4 standard output to a new Fig. 5 showing the weekly averaged FCOS, and in Fig. 6 showing the cumulative FCOS.

[Figure]

The largest departure between the optimized model and observations occurs in July 2014. It is noted that VPD and temperature were both high in the afternoons during this period. Was this a drought or typical variation? This could be an interesting time period to investigate improving the modeling effort. Error attribution would be helpful and/or comparing to another model like SiB4.

The dry conditions in July 2014 were not really drought but dryer than normal (Fig. 1, Fig. S1). We agree with the reviewer that the disagreement between measured and simulated flux data need to be better addressed in the text. We tried to take the disagreement into account in the parameterization so that only 2014 data was used in determining the fitting parameters. However, this resulted only in a better fit in 2014, but a worse fit for all other years. We found that there is not enough of data from periods with high VPD, so that we could do a reliable parameterization. We have added a paragraph discussing the differences in Section 3.4 Parameterization of the net carbonyl sulfide flux "*In April -June 2013 neither VPD nor SWC were significantly higher or lower, respectively, than in other years (Fig. S1). In July 2014 VPD was about twice as high as in 2015-2017, and so FCOS was likely low due to stomatal regulation. In April 2015 SWC is lower than in other years which may similarly explain the low COS flux in that period. In April 2016 neither VPD nor SWC were significantly higher or lower, respectively, than in other years. High VPD or low SWC may also be reflected in the net $CO_2$ eocsystem exchange (NEE). However, NEE values do not differ from other years during these periods of low COS uptake, excluding the highest carbon uptake (lowest NEE), in contradiction to FCOS, in July 2014 (Fig. S1). However, PAR was higher in July 2014 than in any other year, which may explain high carbon uptake. FCOS is more sensitive to the stomatal conductance than the $CO_2$ exchange is, because the changes of the sub-stomatal $CO_2$ concentration are partly buffering the direct effect of the stomatal closure.*

*The local parameterization and the parameterization with SiB4 data are close to each other, while the SiB4 simulation underestimates FCOS especially during summertime and fall (Fig. 5). ".*

Some explanation as to the provenance of the process-informed equations would help a wider readership understand where they came from. For example, the VPD equation looks related to the one mentioned in Medlyn et al., 2011 which itself was based on earlier work. It would be useful to justify why this approach is used instead of Ball-Berry, for example. Also, I have not encountered the variable S before to be able to recognize it as a measure of phenology. All this is not helped by some possible errors in rendering the equations themselves.

We wanted to have the stomatal VPD regulation to be based on stomatal theories rather than empirical studies (such as Ball-Berry) or semi-empirical (Medlyn et al., 2011). Thus we have decided to use the model based on Dewar et al., 2018. Dewar et al. (2018) showed that the function FVPD is predicted theoretically by a variety of stomatal optimization models, and explains observed stomatal responses better than the empirical Ball-Berry model. This also now clarified in Section 2.6: "*The VPD response was based on Dewar et al. (2018), who showed that the FVPD function is predicted theoretically by a variety of stomatal optimization models, and explains observed stomatal responses better than the empirical Ball-Berry model (Ball et al., 1987)."*

We have added explanation about the S parameter in Section 2.3: "*The S parameter has been used in ecosystem research especially to determine the spring recovery (Suni et al., 2003; Pelkonen & Hari 1980) and biosphere modeling in JSBACH (Mäkelä et al., 2016; Mäkelä et al., 2019). This is very useful especially at high latitudes due to variating spring conditions. When temperatures may change by over 10°C in one day and vary both below and above 0°C, the more traditional heat sum*

*is not a good approximation for phenology. The S parameter takes into account the temperature history and sub zero temperatures, which the traditional heat sum does not."*

This analysis does not partition ecosystem fluxes between soil and leaf uptake. Sun et al., 2018 measured and analyzed a substantial soil flux record at this site. While it is difficult to satisfactorily model the soil fluxes here, it is straightforward to average them over seasons and include them as an uncertainty for analyzing the ecosystem OCS fluxes. This may prove to be important since the semi-empirical model is based on leaf interactions alone. Related, examining the daytime/nighttime OCS fluxes could be an important foray into nighttime stomatal conductance if soils and other minor fluxes are taken into account. There are many directions to improve the impact of this study as you suggest in section 3.5. Is there a way that we could use this data and this modeling effort to improve the performance or diagnose problems in SiB4 or another land surface model? Can we use this information to compare the many large-scale regional modeling efforts that have recently come out? Of note, the study by Hu et al., 2021 focusses on the Arctic and Remaud et al., 2021 finds a different pattern than the Ma et al., 2021 study.

We focus on ecosystem net fluxes without partitioning them into components such as canopy exchange or soil exchange. We understand that the parameterization mainly takes into account the canopy processes, but as we have seen from soil flux measurements at Hyytiälä (Sun et al. 2018), the COS soil flux does not have a diurnal pattern and changes only slightly throughout the season (ca. from -3 to -4 pmol m$^{-2}$s$^{-1}$), compared to a high seasonal variation in the net ecosystem COS exchange (ca. from -2 to -22 pmol m$^{-2}$s$^{-1}$). This variation is taken into account in the variation of the parameterization function FS, that takes into account the seasonal variation in phenology.

Based on the reviewer's comments we have added discussion and recommendations for future SiB4 development based on the comparison of the long-term FCOS timeseries in section 3.4:
*"The local parameterization and the parameterization with SiB4 data are close to each other, while the SiB4 simulation underestimates FCOS especially during summertime and fall (Fig. 5). Moreover, the decrease in FCOS that was observed in July-August 2014 due to warm and dry conditions was not simulated by SiB4. Instead, SiB4 simulated larger FCOS than other years (Fig. 5, 6) as a result of higher temperatures. It is likely that SiB4 simulates a too small response of the stomata to dry conditions for ENF specifically. A similar result was found for $CO_2$ fluxes by Smith et al. (2020), a study on the European drought in the summer of 2018. They demonstrated that SiB4 does show a drought response, but especially site observations at ENF ecosystems showed a stronger decline in carbon uptake than SiB4 did. In SiB4, ENF is specifically set to be resilient to droughts by setting a lower limit to soil moisture stress on photosynthesis, which is used to derive the stomatal conductance and thereby connects to COS leaf uptake. The COS flux timeseries provide additional evidence that the lower bound on soil moisture stress in ENF ecosystems should be removed. Regarding the general underestimation of FCOS by SiB4, Kooijmans et al. (2021) also found that underestimations of FCOS at Hyytiälä were consistent with underestimations of GPP estimates. Possible methods to increase these fluxes in SiB4 is to increase the maximum carboxylation rate of RuBisCO that is also used to simulate the carbonic anhydrase activity, relevant for COS uptake. Another approach is to incorporate a $CO_2$ fertilization effect that would increase the aboveground biomass and would increase both fluxes. Research is ongoing to implement an accurate representation of the $CO_2$ fertilization effect in SiB4, at the same time assessing other processes like respiration and water use efficiency. Simulations of COS soil uptake in Hyytiälä are also too low (Kooijmans et al., 2021), which could be improved with more accurate carbonic anhydrase uptake parameters specific to the ENF soil. Several studies also suggested the role of bryophytes (Gimeno et al., 2017) and epiphytes (Rastogi et al., 2018), which may play a role in boreal forests, but which are not specifically included in SiB4. "*

Finally, some additional data does exist in the broader study area. Multiple efforts by Rastogi et al. in 2018 examined fluxes at a site that appears to be in the southwestern corner of Figure 5. It would seem prudent to compare your model performance with an additional observational dataset.

While it is a good idea to compare the larger scale parameterization to other field sites as well, we are not aware of any other measurement site that would be a boreal evergreen needleleaf forest and have longer term ecosystem scale COS flux measurements, that would be comparable to the parameterization. E.g. Rastogi et al., 2018 experiments, while falling in the map we provided, had a short data set (1 month) with only 15 % of flux data available, making daily averages (that would be comparable to the parameterization) highly uncertain.

Minor comments
194-200 and Table 1: This effort might be better put into the supplement for a curious reader. The main modeling work here is not a multivariate linear regression and including this in the main text seems a bit confusing.
It was previously suggested by two reviewers (in a first submission of the manuscript) to move the multivariate analysis from the supplementary material to the main text. Because other reviewers in this submission did not have a comment on this, we decided to keep it in the main text.

201-209: This analysis could be better incorporated into the overall narrative of this study. The wavelet analysis revealed the lack of time lag between OCS uptake through stomata and PAR, which is reassuring but not surprising. Kooijmans et al., 2019 did an excellent job investigating the role of VPD and OCS uptake at this site. It is not obvious what the wavelet analysis achieves here. This is the first time that wavelet analysis is done for a long-term COS flux dataset. The analysis confirms the results of previous studies in the context of a long data set. We have added a sentence describing this to the Abstract and Conclusions, tying the wavelet analysis better to the overall study: *"Wavelet analysis of the ecosystem fluxes confirmed earlier findings from branch-level fluxes at the same site and revealed a 3-hour lag between FCOS and $T_a$ in the daily scale, while no lag between PAR and FCOS was found."*

242: Were there bounds on the possible values for the empirical fits? Since many of the fittings parameters have multiple significant digits, it stands out that b=1000.
Thank you for pointing this out. The parameters indeed had lower and upper bounds, but it was not previously explained in the text. We have now clarified this in the main text: *"Parameters a = −341.81, b = 1000, c = −0.77, and d = 1.03 were optimized using MATLAB's fminsearchbnd function to find the smallest root mean square error of the parameterized FCOS against the measured (non-gap-filled) FCOS. The fminsearchbnd function finds the minimum of a constrained multivariable function using a derivative-free method (D'Errico, 2021). Parameters were given upper and lower limits of a ϵ [5,500], b ϵ [10,1000], c ϵ [-3,3], d ϵ [1,5]. It is to be noted that parameter b is at its upper limit. However, increasing the limit (until infinity) for parameter b only resulted in a higher value for parameter a, without significantly improving the overall fit. Parameter e = 0.18 was fixed before optimizing the other parameters according to a previous study by Peltoniemi et al., (2015), since parameter e is related to ecosystem phenology specific to the site."*

257: Can some of the increase in OCS uptake interannually be attributed to the canopy growing 2 m over this time period?
Good comment! Based on a study of long-term trends (2001-2017) LAI increases yearly 1-2 % in Hyytiälä (Launiainen et al., 2021, in review) depending on the LAI measurement method, so ~ 5-10 % during the whole measurement period (2013-2017) while the cumulative COS flux increased 53.4 % (July-August) or 12% (April-August) from 2013-2017. According to the optical

measurement showed in Fig. 1 the difference between the smallest peak LAI (2014) to the highest peak LAI (2017) was only 4.4 %. The increase in LAI could be one factor, but cannot explain the whole increase in the cumulative COS flux, especially in the summer. In Launiainen et al., (2021) they found - by using a multi-layer ecosystem model – that observed LAI range from 3.2 to 4.5 $m^2m^{-2}$ increased the GPP 7 %, while maintaining a near-constant surface conductance and evapotranspiration. Knowing the small modelled GPP effect and the conservative canopy conductance we argue that in practise the small LAI increase in 2013-2017 had no effect on COS fluxes. In Launiainen et al. (2021) the LAI trend was mostly due to spruce and deciduous understorey getting denser (which the optical measurements could not capture) which brings in some additional uncertainty but does not change the big picture.

As the interannual increase was raised up by the reviewer, we have added discussion about it in section 3.5: "*The parameterized FCOS$_{cum}$ (both using local meteodata and SiB4 meteodata) shows higher total uptake than FCOS measurements for years 2013-2015 but lower uptake for years 2016 and 2017 (Fig. 6). While both the measured and parameterized FCOS$_{cum}$ have a slight decreasing trend during 2013-2017 (uptake increasing), the SiB4 output shows an opposite trend of increasing FCOS$_{cum}$ (decreasing total COS uptake). However, this time series is too short for analysing trends and the observed differences can be explained by differences in environmental drivers.*"
Note that Fig. 6 was also updated to include FCOS$_{cum}$ values also from the parameterizations (local & SiB4) as well as SiB4 output.

This is interesting and important work and I look forward to seeing its evolution.
Mary Whelan

Citations
Hu, Le, Stephen A. Montzka, Aleya Kaushik, Arlyn E. Andrews, Colm Sweeney, John Miller, Ian T. Baker, Scott Denning, Elliott Campbell, Yoichi P. Shiga, Pieter Tans, M. Carolina Siso, Molly Crotwell, Kathryn McKain, Kirk Thoning, Bradley Hall, Isaac Vimont, James W. Elkins, Mary E. Whelan, Parvadha Suntharalingam: COS-derived GPP relationships with temperature and light help explain high-latitude atmospheric CO2 seasonal cycle amplification, Proceedings of the National Academy of Sciences Aug 2021, 118 (33) e2103423118; DOI: 10.1073/pnas.2103423118

Rastogi, B., Berkelhammer, M., Wharton, S., Whelan, M. E., Meinzer, F. C., Noone, D., and Still, C. J.: Ecosystem fluxes of carbonyl sulfide in an old-growth forest: temporaldynamics and responses to diffuse radiation and heat waves, Biogeosciences, 15, 7127–7139, 2018. https://doi.org/10.5194/bg-15-7127-2018

Rastogi, B., Berkelhammer, M., Wharton, S., Whelan, M. E., Itter, M. S., Leen, J. B., Gupta, M. X., Noone, D., and Still, C. J.: Large Uptake of Atmospheric OCS Observed at a Moist Old Growth Forest: Controls and Implications for Carbon Cycle Applications, J. Geophys. Res. Biogeosci., 123, 3424–3438, 2018. https://doi.org/10.1029/2018JG004430

Remaud, M., Chevallier, F., Maignan, F., Belviso, S., Berchet, A., Parouffe, A., Abadie, C., Bacour, C., Lennartz, S., and Peylin, P.: Plant gross primary production, plant respiration and carbonyl sulfide emissions over the globe inferred by atmospheric inverse modelling, Atmos. Chem. Phys. Discuss. [preprint], https://doi.org/10.5194/acp-2021-326, in review, 2021.

**Reviewer #3** – Anonymous
1 GENERAL COMMENTS
This paper analyses the seasonal and interannual variabilities of 5-years biospheric COS fluxes at a site located in a boreal pine forest in Finland. To explain these variability modes, the relationships between the COS biospheric sink and environmental drivers (vapor pressure deficit, light, air temperature) are described. A linear regression model is used to select the main environmental drivers of COS biospheric sink variability. They further develop an empirical model of the COS biospheric sink that is function of PAR, LAI and VPD. The model calibrated with the observations successfully reproduces the observed seasonal and interannual variabilities. Then, they optimize the

parameters of the model using the LAI, VPD, PAR simulated by the SIB4 Land Surface Model along with the observed COS fluxes at the site of interest. The calibrated model is then applied to the evergreen boreal needleleaf forests over the whole northern hemisphere. The total COS biospheric sink is greater than the simulated one by the SIB4 LSM, in agreement with a missing sink inferred by most top-down studies.

Overall, the paper provides a valuable advance in reconciling the bottom-up and top-down COS budget at high latitude. However, the manuscript seems to be written in a rush. The many references in the Supplementary makes it difficult to follow sometimes and some clarifications are needed. In particular, I have a few questions that need to be addressed.

We thank the reviewer for the very insightful and constructive comments. The manuscript was initially intented for a shorter format journal, which is why only the main results ended up in the main text. Since ACP allows for more text in the main text as well, we have now moved all the methodological text in the previous Supplementary material to the main text in the Materials and Methods section.

1. In agreement with reviewer 2, the empirical relations 2-5 call for more explanations of their provenance and their physical meanings in the main text. In the equation 5, it is unclear how the parameter e is fixed. For the observations, how and why was the parameter e fixed before optimizing the other parameters (Page 4, line 107)? For the SIB4 LSM, why is e multiplied by 2.1, the average ratio of Hyytiala and SIB4 LAI data (Page 5 line 2)?

We have clarified the parameterization in Section 2.6 as follows "*Since the ecosystem COS uptake is dominated by the canopy uptake (70% at minimum according to Sun et al., 2018) and is process-wise very close to the CO₂ uptake, we formulated the parameterization as follows:*
*…*
*where a, b, c, d, and e are fitting parameters and four functions are simplified from the corresponding dependencies of the CO₂ uptake on PAR and T_a (S being a function of T_a; see Text S2) (according to Mäkelä et al., 2008), VPD (Dewar et al., 2018), and LAI (Peltoniemi et al., 2015). Parameters a = −341.81, b = 1000, c = −0.77, and d = 1.03 were optimized using MATLAB's fminsearchbnd function to find the smallest root mean square error of the parameterized FCOS against the measured (non-gap-filled) FCOS. The fminsearchbnd function finds the minimum of a constrained multivariable function using a derivative-free method (D'Errico, 2021). Parameters were given upper and lower limits of a ϵ [5,500], b ϵ [10,1000], c ϵ [-3,3], d ϵ [1,5]. It is to be noted that parameter b is at its upper limit. However, increasing the limit (until infinity) for parameter b only resulted in a higher value for parameter a, without significantly improving the overall fit. Parameter e = 0.18 was fixed before optimizing the other parameters according to a previous study by Peltoniemi et al., (2015), since parameter e is related to ecosystem phenology specific to the site. Although recognizing that other more complex formulas with more fitting parameters could provide better correspondence with observations, we desired to keep the parameterization simple for the sake of generic process description: here FPAR describes the stomatal response to PAR and includes all other light-dependent processes, FS the phenology of biochemical reactions, FVPD the stomatal regulation, and FLAI the amount of foliage and canopy light penetration.*"

We have now produced a new figure (Fig. S7) that shows the relation of each parameter functions to the environmental drivers they describe. In equation 5 (now Eq.11) parameter *e* was fixed based on earlier studies on CO2 exchange (Peltoniemi et al., 2015). The reason we believe the parameter should not depend on gas (and should be same for CO2 and COS) is that the FLAI function describes the ecosystem phenology that is not gas-dependent. The gas-dependency is taken into account with the fitting parameters in FPAR and FVPD functions. We clarified the *e* parameter

origin in the text as follows: *"Parameter e = 0.18 was fixed before optimizing the other parameters according to a previous study by Peltoniemi et al., (2015), since parameter e is related to ecosystem phenology specific to the site".*

The LAI data from in-situ measurements and SiB4 differ because they represent different LAI. The in-situ LAI is the all-sided leaf area index, while SiB4 LAI is projected leaf area index. This is now clarified in the text as *"To obtain COS biosphere fluxes for the whole boreal region based on the FCOS observations in Hyytiälä we calibrated the SiB4 PAR, LAI, VPD and $T_a$ for the grid cell where Hyytiälä is located against observations (see Fig. S10). We then applied the parameterization represented in Eq. 7-11 to the whole boreal region (based on the ENF grid cell selection described in the previous paragraph) using the SiB4 meteorological and phenological data.".*

2. This concerns the text around Table 1, where the authors attempt to disentangle the drivers of the COS biospheric sink variability. Given the high non-linearity of the underlying equations 2-5, it is not clear how well the simple multiple linear regression analysis presented in Table 1 is able to capture such highly non-linear interactions. To my opinion, the statistical analysis would require a different approach, such as a decision tree or a neural network in order to deal with the highly non-linear interactions between variables.

It is true that the relations are mainly non-linear, but the linear regressions give some insight to which parameters are important. Often highly non-linear correlations also have higher linear correlation than when there is no correlation at all. Decision trees decide variable importance also based on linear relations, so we don't see it as a very different approach to what we have done with the multivariate linear regressions. Neural networks on the other hand do take into account other types of interactions as well, but it becomes difficult to decide on a good and reliable metric to compare such differently shaped responses to find the most important drivers. Since the regression analysis is not the main focus of the study but only gives some background information to the COS flux variations, we have decided to leave the multivariate regression analysis as is.

3. To upscale the flux observations to evergreen needleleaf forests in the whole boreal region, it would have been more straightforward to use the MODIS products of PAR, LAI along with the surface air temperature, VPD from the MERRA reanalysis. The MODIS-derived LAI and PAR should be more realistic spatial variability than the ones simulated by the LSM. A processed-based alternative would have been to (i) calibrate the parameters of the SIB4 LSM with the observed LAI, PAR, VPD, FCOS at the closest grid point of the eddy covariance site and (ii) to run the newly calibrated LSM to simulate the COS biospheric fluxes over evergreen needleleaf forests in the whole boreal region. Such analysis would have shed light on the specific LSM parameters that need to be calibrated. The use of the SIB4 LSM would be better justified if the authors evaluate the performance of the LSM to reproduce the COS biospheric sink at Hyytiala (for instance Fig.4). Also, the statistical analysis in Table 1 could be also done for the SIB4 LSM and compared with Maignan et al., 2021 who did a similar analysis.

We thank the reviewer for these useful suggestions. For as far as we know, the MODIS LAI product is not available on daily timescale and can be noisy when looking at a single pixel/gridcell. While the information on spatial variability of the MODIS product would have been useful, we choose to keep using the SiB4 LAI for consistency with the SiB4 COS fluxes and for the possibility to obtain LAI from a specific plant functional type (in our case evergreen needleleaf forest). We do agree with the reviewer that calibrating the SiB4 LAI and meteo data (PAR, VPD and T) to in-situ measurements is an appropriate choice. Therefore, we decided to adopt the suggestion of the reviewer to i) calibrate the SiB4 meteodata based on in-situ measurements (plot 2 in response to reviewer #2, Fig. S10 in the revised supplement) and ii) to use the calibrated SiB4 meteodata and

the same parameterization as developed for in-situ data in Hyytiälä. Figures 7 and S11 and numbers in the text have been updated accordingly.

The cumulative COS fluxes from the in-situ parameterization, SiB4 standard output and SiB4 with the simple parameterization at Hyytiälä grid have been added to Fig. 6 (previous Fig. 4) as suggested. We also added a new Fig. 5 showing the weekly averages of measured and parameterized local FCOS, as well as SiB4 parameterization and simulation.

We have added discussion on how the SiB4 parameterization compares to Hyytiälä parameterization and observations to section 3.4: "*The local parameterization and the parameterization with SiB4 data are close to each other, while the SiB4 simulation underestimates FCOS especially during summertime and fall (Fig. 5). Moreover, the decrease in FCOS that was observed in July-August 2014 due to warm and dry conditions was not simulated by SiB4. Instead, SiB4 simulated larger FCOS than other years (Fig. 5, 6) as a result of higher temperatures. It is likely that SiB4 simulates a too small response of the stomata to dry conditions for ENF specifically. "*
and section 3.5: "*The parameterized $FCOS_{cum}$ (both using local meteodata and SiB4 meteodata) shows higher total uptake than FCOS measurements for years 2013-2015 but lower uptake for years 2016 and 2017 (Fig. 6). While both the measured and parameterized $FCOS_{cum}$ have a slight decreasing trend during 2013-2017 (uptake increasing), the SiB4 output shows an opposite trend of increasing $FCOS_{cum}$ (decreasing total COS uptake). However, this time series is too short analysing trends and the observed differences can be explained by differences in environmental drivers.*"

We did not repeat the statistical analysis (that was done for measured FCOS) for SiB4 output, because the intent was to check which drivers are the most important based on measured flux data. However, we think it is a good idea to compare our regression analysis results to those of Maignan et al., (2021) and we have added discussion on that in the main text: "*Maignan et al., (2021) studied the importance of different drivers (PAR, $T_a$, VPD, LAI, SWC) to stomatal conductance in Hyytiälä using random forest models. The three most important drivers for stomatal conductance were (in order) PAR, $T_a$ and VPD. This is well in line with our univariate analysis that ranked temperature and radiation as the most important drivers of FCOS, that is mostly regulated by stomatal conductance. The importance of VPD is larger on longer time-scales.*"

2 SPECIFIC COMMENTS
Page 2, line 41-42: "The terrestrial ... 1360 GgS/y" I would add Remaud et al. (2021) who used a more recent anthropogenic inventory to infer the net biospheric sink over the globe through inverse modelling. The missing sink in the high northern latitudes was also shown in Remaud et al. (2021); Hu et al. (2021).
Reference to Remaud et al., (2021) added (as Hu et al. was not a global study). Remaud et al., (2021) and Hu et al., (2021) have been also added as references elsewhere in the text, when discussing the missing high latitude sink.

Page 3, line 7: "of which is gap-filled ..." The method for gap-filling needs to be explained for the sake of clarity.
Gap-filling method is now explained in the Materials and Methods section (instead of Supplement as previously).

Page 3, line 91: I would add "at 23 m height, where the EC measurements were made".
Corrected as suggested

Page 3, line 95: Where are located the 5 locations?
Added "*representative of the forest floor*" as the five locations are around the measurement site

Page 4, line 106: Here, I would present the parameterization of Carbonyl sulfide fluxes by showing and explaining the equations 1-5.
Corrected as suggested, the parameterization explanation was moved from Results section to Materials and Methods section and the text was clarified (as explained earlier in the response).

Page 6, line 146, "However, the snow ... region." What is the link with the former sentence?
Removed "However,"

Page 6, line 153: "We analyzed ... FCOS" What is the dependence of the CO2 flux to the VPD?
Here we explained how we tried to find a temperature threshold for the commencement of the $CO_2$ and COS uptakes in the spring, but found no such threshold, unlike in a previous study by Suni et al., 2003.

Page 7, line 165: Epiphytes can also significantly take up the atmospheric COS during the night when the soil is wet as shown by Rastogi et al. (2018).
Added this information in the text: "*Mosses can also significantly take up COS during the night when the soil is wet (Rastogi et al., 2018). Nighttime soil uptake in Hyytiälä was ca. -3 pmol $m^{-2}s^{-1}$ (Sun et al., 2018) while the ecosystem scale nighttime uptake in our study is ca. -10 pmol $m^{-2}s^{-1}$. However, we did not measure the contribution from mosses. The nighttime COS uptake in Hyytiälä is thus likely a combination of soil and moss uptake, but also has a larger contribution from the canopy (Kooijmans et al., 2017).* "

Page 7, line 182: What does "unfiltered" data mean? Was the Fig. S6 drawn with the data that have not been gap-filled?
By unfiltered data we meant that the data were not filtered for high PAR values only (as was done in Fig. 2), nor separated into seasons. We have now clarified the text as "*Responses to $T_a$ and VPD were filtered to only include data with PAR > 500 µmol $m^{-2}$ $s^{-1}$ to avoid including radiation related correlation. For the responses without separation between spring and summer periods and without filtering by low PAR, see Fig. S5*". Fig. S5 (previously Fig. S6) was drawn from measured data only (not gap-filled), this is now also clarified in the caption.

Page 7, line 201: The set-up of the wavelet analysis should be explained in the method. Figure S7 should be in the main text.
Corrected as suggested. The wavelet coherence analysis is now explained in the methods section 2.5, instead of the Supplementary material. We have decided to keep the wavelet figure in the Supplementary material, because the other reviewer comments somewhat disagree with this suggestion.

Page 7, lines 194-209: See Maignan et al. (2021) (Fig. 3). Based on the ORCHIDEE LSM, they showed that the internal conductance was mainly driven by Ta. In the afternoon, the internal conductance limits the total conductance and reaches a maximum 3 hours after the peak in stomatal conductance.
Thank you for pointing this out! We have added discussion about this in the text: "*In addition, Maignan et al., (2021) showed that the internal conductance is driven by $T_a$ and limits the total conductance, especially in the afternoon.*"

Page 9, Table 1: I wonder how the result would look like if the measurements were not gap-filled.
The multivariate regression analysis was done based on measured data only, so that e.g. daily averages were calculated only if >50% of measured data existed that day. The regression analysis was not done to gap-filled data, because the gap-filling funtion already introduce responses to environmental parameters. We have clarified this in the text: "*Only measured (non-gap-filled)*

*FCOS data was used for the analysis so that minimum 50 % of data had to exist to calculate e.g. daily averages.*"

Page 10, equations 1-5 : A plot associated with each equation (e.g FPAR as function of PAR) would give a better idea of their contribution to FCOS.
Thank you for this suggestion! We have added such plot to the supplementary material, Fig. S7.

Page 10, line 247: It should be mentioned earlier in Part 2.4 that the parameterization is not applied to the gap-filled data.
We had written "*to find the smallest root mean square error of the parameterized FCOS against the measured FCOS*" but have now added clarification "non-gapfilled" to make clear it is indeed only measured data.

Page 10, line 252: It is worth discussing here the reasons why the empirical model and the observations for July 2014 are in disagreement for July 2014 (Figure 1). Does the empirical model underestimate the sensitivity of the surface fluxes to vapor pressure deficit? Does the net CO2 fluxes exhibit such decrease during the same period?
We agree with the reviewer that the disagreement between measured and simulated flux data in July 2014 need to be better addressed in the text. We tried to take the disagreement into account in the parameterization so that only 2014 data was used in determining the fitting parameters. However, this resulted only in a better fit in 2014, but a worse fit for all other years. We added a figure showing the monthly NEE averages for each year (Fig. S1 d) and a paragraph discussing the differences in Section 3.4: "*In April -June 2013 neither VPD nor SWC were significantly higher or lower, respectively, than in other years (Fig. S1). In July 2014 VPD was about twice as high as in 2015-2017, and so FCOS was likely low due to stomatal regulation. In April 2015 SWC is lower than in other years which may similarly explain the low COS flux in that period. In April 2016 neither VPD nor SWC were significantly higher or lower, respectively, than in other years. High VPD or low SWC may also be reflected in the net $CO_2$ eocsystem exchange (NEE). However, NEE values do not differ from other years during these periods of low COS uptake, excluding the highest carbon uptake (lowest NEE), in contradiction to FCOS, in July 2014 (Fig. S1). However, PAR was higher in July 2014 than in any other year, which may explain high carbon uptake. FCOS is more sensitive to the stomatal conductance than the $CO_2$ exchange is, because the changes of the sub-stomatal $CO_2$ concentration are partly buffering the direct effect of the stomatal closure.*"

Page 11, line 263: "The cum.. respectively." What are the numbers within parenthesis? IS FCOScum computed using the gap-filled measurements? Page 11, lines 265-267: I don't understand the link between the COS sulfur deposition and the Carbonyl sulfide balances and their interannual variation. Please explain.
We will add the sentence: "*We can compare the amount of sulfur deposited by COS with the dominant sulfur compounds.*" We hope this clarifies the link to sulphur deposition. The numbers in the parenthesis represent cumulative fluxes in units gS ha$^{-1}$ while the non-parenthesis values are in units µmol COS m$^{-2}$, as explained in the text. FCOScum is computed using the gap-filled fluxes. For clarity, we have separated the COS and S units of the cumulative fluxes into two sentences: "*The total $FCOS_{cum}$ for the period April–August was −183 ± 41, −130 ± 30, −207 ± 45, −205 ± 45 µmol COS m$^{-2}$ for years 2013, 2014, 2016, and 2017, respectively. Expressed as sulfur uptake the $FCOS_{cum}$ become −58.5 ± 13, −41.6 ± 9.6, −66.4 ± 14.4, −65.5 ± 14.4 gS ha$^{-1}$ for years 2013, 2014, 2016, and 2017, respectively.*

Page 12, Figure 4: I find Figure 4 particularly interesting, showing that the COS biospheric sink increases between 2013-2017 while the average atmospheric COS concentration decreases over the whole northern hemisphere during the same period. Has the length of the growing season been

increasing over the years? It would be interesting to investigate the reasons underlying this increase, for instance by looking at the environmental drivers of FCOS. It would also be interesting to reproduce the Figure 4 for the SIB4 LSM.

Launiainen et al. 2021 (in review) shows that the growing season length has not changed over a period 2001-2017. As pointed out by reviewer #2, increasing LAI on the other hand may have a role in the increasing cumulative FCOS. The LAI has increased in Hyytiälä 1-2 % (Launiainen et al., 2021) so altogether 5-10 % during our measurement period 2013-2017 while the FCOS$_{cum}$ has increased 53 % (July-August) or 12 % (April-August). The increase in LAI could be one factor, but cannot explain the whole increase in the cumulative COS flux, especially in the summer. In Launiainen et al., (2021) they found - by using a multi-layer ecosystem model – that observed LAI range from 3.2 to 4.5 m$^2$m$^{-2}$ increased the GPP 7 %, while maintaining a near-constant surface conductance and evapotranspiration. Knowing the small modelled GPP effect and the conservative canopy conductance we argue that in practise the small LAI increase in 2013-2017 had no effect on COS fluxes. In Launiainen et al. (2021) the LAI trend was mostly due to spruce and deciduous understorey getting denser (which the optical measurements could not capture) which brings in some additional uncertainty but does not change the big picture.

We have reproduced Fig. 6 (previously Fig. 4) as suggested.

Page 12, line 273: "..improve the .. sink estimate." and line 285 "... could help to improve the representation of gross primary production in biospheric models as well.". The paper falls a bit short of really providing a parameterization that could help to improve the LSMs. This is more true that this study highlights a LSM bias which is an underestimation of the simulated biospheric sink in the high latitudes, in agreement with recent inverse modelling studies (Ma et al., 2021; Remaud et al., 2021).

We have added discussion and suggestions for model improvement in section 3.4: *"The local parameterization and the parameterization with SiB4 data are close to each other, while the SiB4 simulation underestimates FCOS especially during summertime and fall (Fig. 5). Moreover, the decrease in FCOS that was observed in July-August 2014 due to warm and dry conditions was not simulated by SiB4. Instead, SiB4 simulated larger FCOS than other years (Fig. 5, 6) as a result of higher temperatures. It is likely that SiB4 simulates a too small response of the stomata to dry conditions for ENF specifically. A similar result was found for $CO_2$ fluxes by Smith et al. (2020), a study on the European drought in the summer of 2018. They demonstrated that SiB4 does show a drought response, but especially site observations at ENF ecosystems showed a stronger decline in carbon uptake than SiB4 did. In SiB4, ENF is specifically set to be resilient to droughts by setting a lower limit to soil moisture stress on photosynthesis, which is used to derive the stomatal conductance and thereby connects to COS leaf uptake. The COS flux timeseries provide additional evidence that the lower bound on soil moisture stress in ENF ecosystems should be removed. Regarding the general underestimation of FCOS by SiB4, Kooijmans et al. (2021) also found that underestimations of FCOS at Hyytiälä were consistent with underestimations of GPP estimates. Possible methods to increase these fluxes in SiB4 is to increase the maximum carboxylation rate of RuBisCO that is also used to simulate the carbonic anhydrase activity, relevant for COS uptake. Another approach is to incorporate a $CO_2$ fertilization effect that would increase the aboveground biomass and would increase both fluxes. Research is ongoing to implement an accurate representation of the $CO_2$ fertilization effect in SiB4, at the same time assessing other processes like respiration and water use efficiency. Simulations of COS soil uptake in Hyytiälä are also too low (Kooijmans et al., 2021), which could be improved with more accurate carbonic anhydrase uptake parameters specific to the ENF soil. Several studies also suggested the role of bryophytes (Gimeno et al., 2017) and epiphytes (Rastogi et al., 2018), which may play a role in boreal forests, but which are not specifically included in SiB4. "*

Page 12, line 273: I would rather say: "with the recent top-down studies of the COS atmospheric budget (Ma et al., 2021; Remaud et al., 2021; Hu et al., 2021)"

Corrected as suggested.

**3 References**

Hu, L., Montzka, S. A., Kaushik, A., Andrews, A. E., Sweeney, C., Miller, J., Baker, I. T., Denning, S., Campbell, E., Shiga, Y. P., Tans, P., Siso, M. C., Crotwell, M., McKain, K., Thoning, K., Hall, B., Vimont, I., Elkins, J. W., Whelan, M. E., and Suntharalingam, P.: COS-derived GPP relationships with temperature and light help explain high-latitude atmospheric CO2 seasonal cycle amplification, Proceedings of the National Academy of Sciences, 118, https://doi.org/10.1073/pnas.2103423118, https://www.pnas.org/content/118/33/e2103423118, publisher: National Academy of Sciences Section: Physical Sciences, 2021.

Ma, J., Kooijmans, L. M. J., Cho, A., Montzka, S. A., Glatthor, N., Worden, J. R., Kuai, L., Atlas, E. L., and Krol, M. C.: Inverse modelling of carbonyl sulfide: implementation, evaluation and implications for the global budget, Atmospheric Chemistry and Physics, 21, 3507–3529, https://doi.org/10.5194/acp-21-3507-2021, https://acp.copernicus.org/articles/21/3507/2021/, publisher: Copernicus GmbH, 2021.

Maignan, F., Abadie, C., Remaud, M., Kooijmans, L. M. J., Kohonen, K.-M., Commane, R., Wehr, R., Campbell, J. E., Belviso, S., Montzka, S. A., Raoult, N., Seibt, U., Shiga, Y. P., Vuichard, N., Whelan, M. E., and Peylin, P.: Carbonyl sulfide: comparing a mechanistic representation of the vegetation uptake in a land surface model and the leaf relative uptake approach, Biogeosciences, 18, 2917–2955, https://doi.org/10.5194/bg-18-2917-2021, publisher: Copernicus GmbH, 2021.

Rastogi, B., Berkelhammer, M., Wharton, S., Whelan, M. E., Itter, M. S., Leen, J. B., Gupta, M. X., Noone, D., and Still, C. J.: Large Uptake of Atmospheric OCS Observed at a Moist Old Growth Forest: Controls and Implications for Carbon Cycle Applications, Journal of Geophysical Research: Biogeosciences, 123, 3424–3438, https://doi.org/https://doi.org/10.1029/2018JG004430, 2018.

Remaud, M., Chevallier, F., Maignan, F., Belviso, S., Berchet, A., Parouffe, A., Abadie, C., Bacour, C., Lennartz, S., and Peylin, P.: Plant gross primary production, plant respiration and carbonyl sulfide emissions over the globe inferred by atmospheric inverse modelling, Atmospheric Chemistry and Physics Discussions, pp. 1–43, https://doi.org/10.5194/acp-2021-326, publisher: Copernicus GmbH, 2021.

**References**

Dewar, R., Mauranen, A., Mäkelä, A., Hölttä, T., Medlyn, B., & Vesala, T. (2018). New insights into the covariation of stomatal, mesophyll and hydraulic conductances from optimization models incorporating nonstomatal limitations to photosynthesis. New Phytologist, 217(2), 571-585.

Kooijmans, L. M., Sun, W., Aalto, J., Erkkilä, K. M., Maseyk, K., Seibt, U., Vesala, T., Mammarella, I. & Chen, H. (2019). Influences of light and humidity on carbonyl sulfide-based estimates of photosynthesis. Proceedings of the National Academy of Sciences, 116(7), 2470-2475.

Launiainen, S., Katul, G.G., Leppä, K., Kolari, P., Aslan, P., Grönholm, T., Korhonen, L., Mammarella, I. and Vesala, T. (2021). Does growing atmospheric CO2 explain increasing carbon sink in a boreal coniferous forest? Global Change Biology, in review

Mäkelä, A., Pulkkinen, M., Kolari, P., Lagergren, F., Berbigier, P., Lindroth, A., Loustau, D., Nikinmaa, E., Vesala, T., and Hari, P. 2008. Developing an empirical model of stand GPP with the LUE approach: analysis of eddy covariance data at five contrasting conifer sites in Europe. Global Change Biology, 14, 92-108.

Mäkelä, J., Susiluoto, J., Markkanen, T., Aurela, M., Järvinen, H., Mammarella, I., ... & Aalto, T. (2016). Constraining ecosystem model with adaptive Metropolis algorithm using boreal forest site eddy covariance measurements. *Nonlinear processes in geophysics, 23*(6), 447-465.

Mäkelä, J., Knauer, J., Aurela, M., Black, A., Heimann, M., Kobayashi, H., ... & Aalto, T. (2019). Parameter calibration and stomatal conductance formulation comparison for boreal forests with adaptive population importance sampler in the land surface model JSBACH. *Geoscientific Model Development, 12*(9), 4075-4098.

Pelkonen, P., & Hari, P. (1980). The dependence of the springtime recovery of CO2 uptake in Scots pine on temperature and internal factors. Flora, 169(5), 398-404.

Peltoniemi, M., Pulkkinen M., Aurela M., Pumpanen J., Kolari P. & Mäkelä, A. 2015: a semi-empirical model of boreal-forest gross primary production, evapotranspiration, and soil water — calibration and sensitivity analysis. Boreal Env. Res. 20: 151–171

Sun, W., Kooijmans, L. M., Maseyk, K., Chen, H., Mammarella, I., Vesala, T., Levula, J., Keskinen, H. & Seibt, U. (2018). Soil fluxes of carbonyl sulfide (COS), carbon monoxide, and carbon dioxide in a boreal forest in southern Finland. Atmospheric Chemistry and Physics, 18(2), 1363-1378.

Suni, T., Berninger, F., Vesala, T., Markkanen, T., Hari, P., Mäkelä, A., Ilvesniemi, H., Hänninen, H., Nikinmaa, E., Huttula, T., Laurila, T., Aurela, M., Grelle, A., Lindroth, A., Arneth, A., Shibistova, O. & Lloyd, J. (2003). Air temperature triggers the recovery of evergreen boreal forest photosynthesis in spring. Global change biology, 9(10), 1410-1426.

---

## Referee Report (RR1)

**Review : Long-term fluxes of carbonyl sulfide and their seasonality and interannual variability in a boreal forest**

I find that the manuscript has been significantly improved. The authors replied to my comments in an appropriate manner and followed my main suggestions. They added more explanations in the "materials and methods" section and performed additional evaluations of the COS fluxes simulated by the SIB4 Land Surface Model against measurements at Hyytiala. There are still some minor points that need to be clarified before publication.

Page 4, line 100: Add a reference as for the provenance of the equation 1.

Page 5, equation 6: Replace $T$ by $T_a$.

Page 6, section 2.6 : Add a sentence precising that the year 2014 was not used in determining fitting parameters.

Page 7, line 197: Consider adding a sentence saying that e is believed not to be gas dependent.

Page 7, section 2.7 : Although the authors added some clarifications in the review regarding the e parameter, they do not appear in the manuscript! Add a few sentences explaining the reasons why e is equal to 2.1. "The in-situ LAI is the all-sided lead area index, while SIB4 LAI is projected leaf area index. For this reason, e in SIB4 is fixed to 2.1".

Page 8, lines 220-224: The authors calibrate now the SIB4 meteodata based on in situ measurements but the calibration method is not mentioned clearly in the manuscript. Add a sentence explaining how the SIB4 meteodata are calibrated with a reference to Figure S10.

Page 10, legend of the Figure 2 : "show daily gap-filled averages (see Text S1)" Where is the Text S1?

Page 11, line 297: "avoid including radiation related-correlation". Explain why there is a radiation related correlation.

Page 14 ,Table 1: Given the high non-linearity underlying the equations 7-11, the statistical analysis would require a random forest approach as done in Maignan et al., 2021 to deal with the high non linear interactions between variables.

---

## Author Response (AR2)

Reviewer comments in black
Author response in blue
*Text in the revised manuscript in italic*

**Reviewer #1**, Jürgen Kesselmeier

The paper was greatly improved. The reader can now reach an overview about environmental factors affecting physiological background for COS-uptake much easier. The content as removed from the supplement and added to the manuscript helps to follow the description and discussion. The additional figures are fine. I have only one minor remark, which may be regarded as a technical correction. The authors discuss the role of other sinks than trees within the boreal ecosystem. However, when regarding the potential role of soils and cryptograms they should not mention "mosses" only. The term "cryptograms" as mentioned in my former review comprises algae, lichens, mosses, and ferns. Furthermore, lichens can provide a large biomass in boreal regions and they can easily reach an ecosystem sink strength of soils (Kuhn et al. 1999, Atmospheric Environment 33, 995-1008). I propose to use the term cryptograms or to write "mosses and other cryptograms".

We thank the reviewer for pointing this out and apologise for the inaccuracy. We have now revised the text as follows: "*Mosses and other cryptogams can also significantly take up COS during the night when the soil is wet (Rastogi et al., 2018). Nighttime soil uptake in Hyytiälä was ca. -3 pmol $m^{-2}s^{-1}$ (Sun et al., 2018) while the ecosystem scale nighttime uptake in our study is ca. -10 pmol $m^{-2}s^{-1}$. However, we did not measure the contribution from cryptogams. The nighttime COS uptake in Hyytiälä is thus likely a combination of soil and cryptogam uptake, but also has a larger contribution from the canopy (Kooijmans et al., 2017).*"

**Reviewer #2**
I find that the manuscript has been significantly improved. The authors replied to my comments in an appropriate manner and followed my main suggestions. They added more explanations in the "materials and methods" section and performed additional evaluations of the COS fluxes simulated by the SIB4 Land Surface Model against measurements at Hyytiala. There are still some minor points that need to be clarified before publication.

Page 4, line 100: Add a reference as for the provenance of the equation 1.
Reference to Kohonen et al., (2020) added.

Page 5, equation 6: Replace T by $T_a$.
Corrected as suggested.

Page 6, section 2.6 : Add a sentence precising that the year 2014 was not used in determining fitting parameters.
Year 2014 was also used in determining the fitting parameters.

Page 7, line 197: Consider adding a sentence saying that *e* is believed not to be gas dependent.
Corrected as suggested: "*Parameter e = 0.18 was fixed before optimizing the other parameters according to a previous study by Peltoniemi et al., (2015), since parameter e is related to ecosystem phenology specific to the site, and is believed not to be gas dependent.*"

Page 7, section 2.7 : Although the authors added some clarifications in the review regarding the e parameter, they do not appear in the manuscript! Add a few sentences explaining the reasons why e

is equal to 2.1. "The in-situ LAI is the all-sided lead area index, while SIB4 LAI is projected leaf area index. For this reason, e in SIB4 is fixed to 2.1".
We apologize for missing to add this clarification also to the manuscript, now corrected.

Page 8, lines 220-224: The authors calibrate now the SIB4 meteodata based on in situ measurements but the calibration method is not mentioned clearly in the manuscript. Add a sentence explaining how the SIB4 meteodata are calibrated with a reference to Figure S10.
Corrected as suggested: *"To obtain COS biosphere fluxes for the whole boreal region based on the FCOS observations in Hyytiälä we calibrated the SiB4 PAR, LAI, VPD and $T_a$ for the grid cell where Hyytiälä is located against observations. The obtained calibration is shown in Fig. S10. The in-situ LAI is the all-sided leaf area index, while SiB4 LAI is projected leaf area index, which explains the large difference between the two LAI data. We then applied the parameterization represented in Eqs. 7-11 to the whole boreal region (based on the ENF grid cell selection described in the previous paragraph) using the SiB4 meteorological and phenological data."*

Page 10, legend of the Figure 2 : "show daily gap-filled averages (see Text S1)" Where is the Text S1?
Thank you for spotting this, it was an old reference. Corrected the reference now as "*(see Sect. 2.2)*".

Page 11, line 297: "avoid including radiation related-correlation". Explain why there is a radiation related correlation.
Added clarification "*since VPD and $T_a$ are highly intercorrelated with PAR.*"

Page 14 ,Table 1: Given the high non-linearity underlying the equations 7-11, the statistical analysis would require a random forest approach as done in Maignan et al., 2021 to deal with the high non linear interactions between variables.
As explained in the previous author response, the linear regressions give some insight to which parameters are important despite non-linearity of some of the interactions. Often highly non-linear correlations also have higher linear correlation than when there is no correlation at all. Since the regression analysis is not the main focus of the study but only gives some background information to the COS flux variations, we have decided to leave the multivariate regression analysis as is. However, we have now added a sentence discussing this: *"While some of the interactions are non-linear, as seen from Fig. 3 and Eqs. (8-11), the linear regression analysis still provides information on the relative importance of the environmental variables, as non-linear correlations usually have a high linear correlation as well. "*

**Reviewer #3**, Mary Whelan

This manuscript presents the first very long term record of carbonyl sulfide (OCS) eddy flux covariance over any ecosystem. Additionally, measurements of OCS exchange in the boreal region are nearly as rare. Even without any analysis, the dataset is valuable to our scientific community. That said, the analysis performed here is the first step of many. A typical motivation for measuring OCS over ecosystems is to reveal new information about the carbon cycle which can be in turn used to constrain the representation of land carbon uptake in land surface models, mentioned in the introduction. The analysis here incorporates the role that stomatal conductance and leaf-affecting parameters play in OCS uptake by vegetation; however, the data is not brought back around to compare to CO2 fluxes. I hope to see this in a future effort!

In the response to my earlier review, you note that deriving stomatal conductance from OCS measurements is "a very difficult task to do from EC" and requires its own paper. Rick Wehr may

have already written this paper in 2017. As far as I can tell, the most difficult part of applying Wehr et al., (2017) approach here is coming up with a reasonable estimate of mesophyll conductance for Scots Pine, which has experienced recent advances (see Stangl et al. 2021). Wehr and Saleska (2021) have also developed an improved method for estimating stomatal conductance from CO2 EC measurements that can be compared to OCS-based estimates.

Thank you for your continued effort in improving this manuscript.

Mary Whelan

References

Stangl, Z.R., Tarvainen, L., Wallin, G. and Marshall, J.D. (2022), Limits to photosynthesis: seasonal shifts in supply and demand for CO2 in Scots pine. New Phytol, 233: 1108-1120. https://doi.org/10.1111/nph.17856

Wehr, R., Saleska, S. (2021) Calculating canopy stomatal conductance from eddy covariance measurements, in light of th energy budget closure problem. Biogeosciences, 18: 13–24. https://doi.org/10.5194/bg-18-13-2021

We thank the reviewer for these comments. This is indeed a first step of many with this data set, that can be used in a multitude of different analyses in the future, not possible to fit in one paper.